# Flexible tungsten disulfide superstructure engineering for efficient alkaline hydrogen evolution in anion exchange membrane water electrolysers

Lingbin Xie[1,2], Longlu Wang ®[1] ✉, Xia Liu[3], Jianmei Chen[1], Xixing Wen[1], Weiwei Zhao[2], Shujuan Liu ®[2] ✉ & Qiang Zhao ®[1,2] ✉

Anion exchange membrane (AEM) water electrolysis employing non-precious metal electrocatalysts is a promising strategy for achieving sustainable hydrogen production. However, it still suffers from many challenges, including sluggish alkaline hydrogen evolution reaction (HER) kinetics, insufficient activity and limited lifetime of non-precious metal electrocatalysts for ampere-level-current-density alkaline HER. Here, we report an efficient alkaline HER strategy at industrial-level current density wherein a flexible $WS_2$ super-structure is designed to serve as the cathode catalyst for AEM water electrolysis. The superstructure features bond-free van der Waals interaction among the low Young's modulus nanosheets to ensure excellent mechanical flexibility, as well as a stepped edge defect structure of nanosheets to realize high catalytic activity and a favorable reaction interface micro-environment. The unique flexible $WS_2$ superstructure can effectively withstand the impact of high-density gas-liquid exchanges and facilitate mass transfer, endowing excellent long-term durability under industrial-scale current density. An AEM electrolyser containing this catalyst at the cathode exhibits a cell voltage of 1.70 V to deliver a constant catalytic current density of $1\,A\,cm^{-2}$ over 1000 h with a negligible decay rate of $9.67\,\mu V\,h^{-1}$.

Developing sustainable energy conversion and storage technologies is essential to mitigate the growing threat of global pollution and overcome fossil fuel depletion. Hydrogen, as a carbon-free energy source with high energy density, is considered one of the ultimate clean and safe energy carriers to reduce the use of fossil fuels and alleviate environmental concerns[1,2]. Low-temperature water electrolysis using renewable electricity is a promising approach to sustainable hydrogen production on a large scale[3–6]. Anion-exchange membrane water electrolysis (AEMWE) combines the advantages of alkaline water electrolysis and proton exchange membrane water electrolysis, and is attracting increasing attention for its potential to realize low-cost and high-performance hydrogen production[7]. However, AEM electrolysis technology is still at a fledgeless stage of development. Considerable effort is required to improve its performance to a level close to that of state-of-the-art proton exchange membrane electrolyzer. For efficient $H_2$ production in AEM electrolyzer, it is vital for the community to

[1]College of Electronic and Optical Engineering & College of Flexible Electronics (Future Technology), State Key Laboratory of Organic Electronics and Information Displays & Jiangsu Key Laboratory for Biosensors, Nanjing University of Posts and Telecommunications, 9 Wenyuan Road, Nanjing 210023, PR China. [2]Institute of Advanced Materials (IAM) & Institute of Flexible Electronics (Future Technology), Nanjing University of Posts and Telecommunications, 9 Wenyuan Road, Nanjing 210023, PR China. [3]College of Chemistry and Chemical Engineering, Qingdao University, Qingdao 266071 Shandong, PR China. ✉e-mail: wanglonglu@njupt.edu.cn; iamsjliu@njupt.edu.cn; iamqzhao@njupt.edu.cn

undertake the essential task of developing active and stable non-precious metal electrocatalysts for the alkaline hydrogen evolution reaction (HER). These catalysts must meet the specific demands of industrial AEMWE while exhibiting low overpotential and prolonged lifetime at an ampere-level current density[8–10].

To date, considerable achievements have been made in the search for high-performance non-precious metal electrocatalysts. It has been reported that the impressive catalytic performance of some non-precious metal electrocatalysts has surpassed that of noble metal benchmarks[11–18]. Nevertheless, it remains a great challenge to develop non-precious metal electrocatalysts that could operate efficiently and stably for long periods of time under industrial hydrogen production conditions (e.g., high temperature (50–90 °C), large current density, and highly concentrated electrolyte). Transition metal dichalcogenides (TMDs) with layered structures have impressive catalytic properties and huge potential for industrial application due to their low cost, high tunability, and excellent electrical conductivity[19–23]. The current limitations of TMD-based catalysts in AEMWE lie in their inferior ability of water dissociation and the deactivation (poisoning) of catalytic active sites during prolonged operation at industrial-level current density, as well as the significantly reduced lifetime attributed to mechanical abrasion resulting from high-density gas-liquid exchanges.

Tungsten disulfide ($WS_2$), as a typical representative two-dimensional (2D) TMDs, is widely regarded as a promising non-precious metal HER catalyst due to its high intrinsic activity at edge sites[22–24]. However, despite extensive efforts in recent years, its high HER activity remains confined to acidic media, with kinetics becoming quite sluggish in more practically viable alkaline environments[22–27]. In addition, during the HER process, 2D-$WS_2$ nanostructures tend to form aggregated materials due to van der Waals (VDW) forces, leading to incomplete exposure of edge sites and consequently diminishing catalytic performance[28,29]. More importantly, 2D-$WS_2$ nanostructures face great challenges in withstanding the mechanical abrasion induced by high-density gas-liquid exchanges, particularly at high current densities (amperage levels), to meet the demands of practical AEM electrolyzer applications[8,17,18]. Some studies have found that designing three-dimensional (3D) $WS_2$ structures could significantly maximize the exposure of active edges, improve reactant accessibility, and enhance mechanical stability, which is an effective strategy for achieving high-level HER performance[30–33]. Nevertheless, until now, the development of $WS_2$-based cathodic electrocatalysts suitable for efficient and long-term hydrogen production in practical AEM electrolyzer has not yet been realized.

2D ultrathin sheets possessing an elastic continuum exhibit excellent mechanical strength and flexibility. This remarkable mechanical property can be further utilized to confer extraordinary mechanical flexibility upon their macroscopic assemblies through the rational design of their structure and sheet interfaces[34–36]. More importantly, the macroscopic assemblies endowed with remarkable mechanical flexibility have the capability to withstand the impact of high-density gas-liquid exchanges by releasing stress through mechanical motion. This, in turn, enables the establishment of a dynamic local micro-environment to prevent deactivation of the active centers while facilitating efficient mass transport of feedstocks and products. Thus, we envision the development of emerging catalysts endowed with both high activity and excellent mechanical flexibility for AEMWE through the elaborated design of the catalytic system with a specifically tailored 3D structure, employing a $WS_2$ nanosheet as the basic unit. Thus, we envision the development of emerging catalysts endowed with both high activity and excellent mechanical flexibility for AEMWE through the elaborated design of the catalytic system with a specifically tailored structure, employing a $WS_2$ nanosheet as the basic unit.

Herein, we report a flexible $WS_2$ superstructure as the cathode catalyst for a practical AEM electrolyzer, which could withstand the impact of high-density gas-liquid exchanges and facilitate mass transport at industrial current density. The $WS_2$ superstructure exhibits excellent HER performance with low overpotentials at industrial-level current densities (e.g., 205 mV@500 mA cm$^{-2}$ and 264 mV@1000 mA cm$^{-2}$) in a three-electrode system, even superior to that of the commercial Pt/C catalyst (e.g., 269 mV@500 mA cm$^{-2}$ and 429 mV@1000 mA cm$^{-2}$). Moreover, a practical AEM electrolyzer catalyzed by $WS_2$ superstructure electrocatalyst achieves alkaline electrolyte electrolysis of 1 A cm$^{-2}$ at 1.70 V and demonstrates a small degradation rate of 9.67 μV h$^{-1}$ during 1000 h of operation at 1 A cm$^{-2}$. This work presents a wonderful flexible superstructure for efficient alkaline hydrogen evolution in AEM electrolyzer, which provides a viable route for the development of industrially applicable gas-involved electrocatalysts.

## Results

### Designing $WS_2$ superstructure with mechanical flexibility

Elaborated design of the basic units composing the macro-assemblies is crucial for the creation of an unusual electrocatalyst that demonstrates high mechanical match and adaptability in AEMWE under industrial-scale current density. The Young's modulus, a fundamental physical quantity employed to characterize a solid material's capacity to resist deformation, represents a crucial parameter when evaluating the mechanical properties of materials[37–41]. To quantify the effect of Young's modulus on the mechanical properties of lamellar materials, we used a simplified sheet model to evaluate the strain required to form a surface topography with a given degree of curvature based on finite element simulations (Fig. 1a). For example, a lamellar material with Young's modulus of 10 GPa requires maximum stress of approximately 0.1 MPa to achieve an interfacial deformation of 20 nm for a given thickness of 10 nm (Fig. 1b). Similarly, we can also simulate the maximum stress that can be achieved for a given 0.1 MPa stress for a laminate with different Young's modulus and thickness (Fig. 1c). These analyses emphasize that the maximum amount of deformation that can be achieved at the bending interface of 2D sheet is proportional to the magnitude of the stresses applied and inversely proportional to Young's modulus and thickness of the sheet.

2D materials generally exhibit in-plane covalent bonding and interlayer VDW interactions. Although the latter is much weaker than the former, the accumulated forces over the large contact area are still too strong for free interlayer sliding motion, leading to a global brittleness in layered solids[42–44]. It has been reported that the Young's modulus of certain 2D materials (such as graphene, α-PbO, $MoS_2$, and $WS_2$) decreases with increasing sheet thickness due to interlayer sliding caused by particular weak interlayer interactions[37,45–47]. More importantly, through the expansion of the layer spacing of 2D sheets by molecular intercalation, the interlayer VDW interactions experience a rapid decay, consequently further enabling the activation of relative motion between the sheets (Fig. 1d). In this particular state, sheet-layer macro-assemblies demonstrate excellent mechanical flexibility in response to tensile, bending, and shear stress effects.

In our designed 3D $WS_2$ superstructure (Fig. 1e), the dangling bond-free nanosheets are staggered with each other in a direction-dependent manner, ensuring excellent charge transport between sheets with minimal interfacial trapping states. Simultaneously, the $WS_2$ superstructure, which is composed of thin sheets and features an expanded distance between layers as well as a plentiful number of exposed edge sites, also offers a greater abundance of available catalytic active sites for electrochemical reactions. Owing to the bond-free VDW interactions between the low Young's modulus nanosheets, the superstructure offers the natural mechanical match required for industrially AEM-relevant high current density and continuous dense

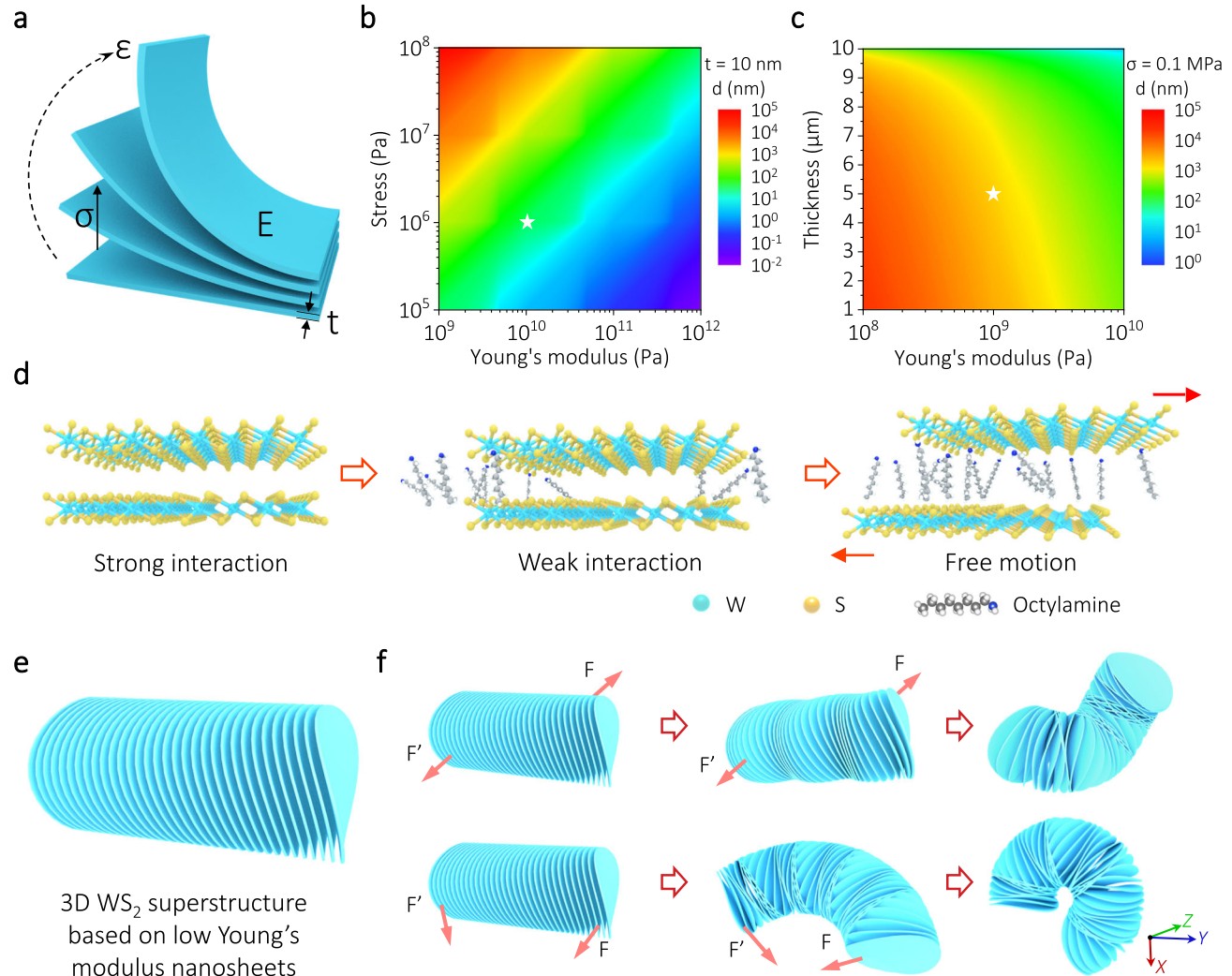

**Fig. 1 | Design and conceptual illustration of flexible WS₂ superstructure.** **a** Diagram of the lamellar model and relevant parameters. **b** Contour maps showing the relationship between Young's modulus, stress, and strain at a thickness of 10 nm. **c** Contour maps showing the relationship between Young's modulus, thickness, and strain at a contact pressure of 0.1 MPa. **d** Schematic diagram of interlayer expansion of 2D layered material. **e** Schematic diagram of WS₂ superstructure based on low Young's modulus nanosheets. **f** Dynamic deformation of flexible WS₂ superstructure in response to shear forces.

gas evolution. A 3D geometric model was engineered to elucidate the dynamic evolution of the sheet macro-assemblies under stress (Fig. 1f). Remarkably, in the event of deformation, the nanosheets with the bond-free VDW interfaces are capable of sliding or rotating relative to each other. This feature is instrumental in accommodating local tension or compressive stresses to ensure the structural and functional integrity of the macro-assembly interfaces and electrical conductance networks (Supplementary Movie 1), which is vital for achieving efficient and robust hydrogen production in practical AEM electrolyzers.

### Structure characterizations and HER mechanism analyses

The WS₂ superstructure was synthesized via an organic solution-phase modulated solvothermal method with WCl₆ and S powder as precursors (see details in "Methods"), as depicted in Supplementary Fig. 1. The optimal synthesis conditions for this method were 200 °C and 48 h, and the method has good repeatability and is feasible for scalable preparation (Supplementary Fig. 2).

The morphological characteristics of the synthesized samples were characterized by scanning electron microscopy (SEM), transmission electron microscopy (TEM), and high-angle annular dark field scanning transmission electron microscopy (HAADF-STEM).

Figure 2a, b shows that the obtained samples are caterpillar-like and comprised of staggered and vertically aligned nanosheets with lateral sizes ranging from 300 to 500 nm. Such 3D nanostructure can exhibit great elastic deformation capability and excellent mechanical flexibility to withstand large strain, which is favorable for efficient collection and conversion of mechanical vibration energy. Supplementary Fig. 3 shows that the WS₂ superstructure is mechanically flexible enough to be twined, bent, and twisted into various shapes during the synthesis process. The HAADF-STEM image and the corresponding energy-dispersive X-ray spectroscopy (EDS) elemental mappings demonstrate the uniform distribution of W and S elements in the sample (Fig. 2c and Supplementary Fig. 4). X-ray diffraction (XRD) analysis was performed on the as-prepared sample to further determine the crystal structure. As shown in Fig. 2d, the (002) diffraction peak of the WS₂ superstructure shifted to 8.9° compared with the 1T-WS₂ and 2H-WS₂ samples, corresponding to an expanded layer spacing of 0.98 nm. The peak at $2\theta \approx 17.6°$ indicates an additional separation of 5.0–5.2 Å between the layers. The Fourier transform infrared (FTIR) study confirmed that the interlayer expansion is due to the intercalation of octylamine molecules during the solvothermal synthesis (Supplementary Fig. 5). Additionally, the peak of (002) crystal plane

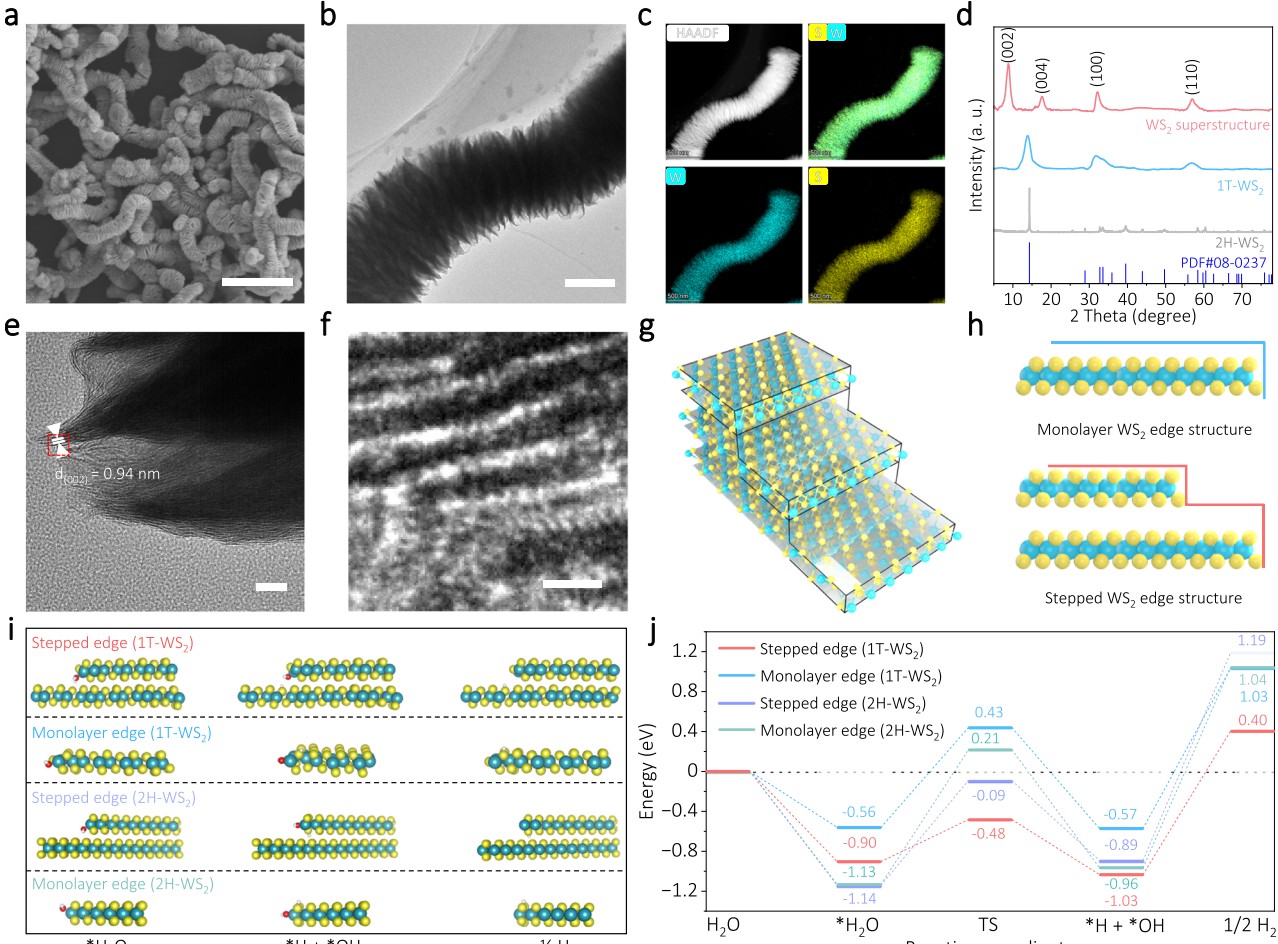

**Fig. 2 | Structure characterization and HER mechanism analyses of the WS₂ superstructure. a** SEM image of the as-prepared WS₂ superstructure. Scale bar, 2 μm. **b** TEM image of the WS₂ superstructure. Scale bar, 200 nm. **c** HAADF-STEM and corresponding elemental mappings of the WS₂ superstructure. **d** XRD patterns of the WS₂ superstructure, 1T-WS₂, and 2H-WS₂ samples. PDF#08-0237 was used as a reference. **e** HRTEM image of the WS₂ superstructure. Scale bar, 10 nm.

**f** Magnified HRTEM image from the remarked areas in (**e**). **g, h** Schematic illustration of the stepped edge defect structures of WS₂ superstructure. **i, j** Schematics and free energy diagram for the alkaline HER of stepped edge (1T-WS₂), monolayer edge (1T-WS₂), stepped edge (2H-WS₂), and monolayer edge (2H-WS₂). The spheres in cyan, yellow, red, and white represent W, S, O, and H atoms, respectively.

has the strongest peak intensity, indicating that the WS₂ superstructure is c-axis extended structure consisting of multilayered S-W-S layers stacked in an ordered manner. X-ray photoelectron spectroscopy (XPS) and Raman analysis further reveal the crystallinity and stoichiometry of the WS₂ superstructure (Supplementary Figs. 6 and 7). As shown in Supplementary Fig. 6a, double peaks located at 31.8 eV and 33.8 eV are ascribed to the core levels of W $4f_{7/2}$ and W $4f_{5/2}$ of 1T phase WS₂ in the sample, respectively. Two peaks of WS₂ superstructure at 32.7 eV (W $4f_{7/2}$) and 34.7 eV (W $4f_{5/2}$) are the characteristics of W for 2H phase WS₂[22,23]. Phase percentages were calculated by peak area ratios of W $4f$ and S $2p$ regions using the deconvolution method[23]. The relative ratio of 1T phase and 2H phase occupies 82.1% and 17.9% in the WS₂ superstructure sample, respectively. Similar results were acquired from the research of S $2p$ core level spectra (Supplementary Fig. 6b). These results revealed that WS₂ existed in hybrid 2H and 1T structures in the as-prepared superstructure. Identical conclusion was obtained in the Raman spectroscopy analysis (Supplementary Fig. 7). The HAADF-STEM image in Supplementary Fig. 8 clearly shows the staggered and vertical alignment between layers.

It is worth noting that the crystal fringes along the edge are stepped for the WS₂ superstructure sample (Fig. 2e, f). The clear lattice fringes with d-spacings of 0.52 and 0.94 nm were observed in the

stepped edge surface-terminated structure (Supplementary Fig. 9). This result is in good consistent with the above XRD analysis. Such edge-stepped defect structure has been reported in MoS₂ and demonstrated to possess abundant catalytic active sites as well as suitable Gibbs free energy ($\Delta G_{H^*}$) for acidic HER[48,49]. Given the compatibility between alkaline HER and industrial AEMWE[50], we employed first principles to explore the potential of WS₂ superstructure featuring abundant unsaturated edge-stepped defects as highly effective non-precious metal electrocatalysts for alkaline HER. Figure 2g, h illustrates the edge-stepped defect structural model constructed based on the observed structural features. It is well known that the reaction kinetics steps of alkaline HER are more complex in comparison to HER under acidic conditions, particularly in terms of the steps involved in the adsorption and dissociation of H₂O[51,52]. The dissociation of *H₂O into *H/*OH on the edge-stepped defect structure exhibits a lower energy barrier in contrast to the monolayer- and bilayer-edge structure (Fig. 2i, j and Supplementary Fig. 10). Additionally, the H* species that are able to adsorb on the surface of S atoms can undergo rapid recombination, resulting in the efficient generation of H₂. The edge-stepped defect structure not only remarkably accelerates the process of water dissociation but also optimizes the hydrogen adsorption free energy on the S atoms located at the edges. This enables the seamless integration of water dissociation with the adsorption and desorption

processes of hydrogen on the WS$_2$ superstructure, which would ultimately yield excellent alkaline HER performance. All atomic coordinates of the optimized computational models were provided in Supplementary Data 1. Moreover, molecular dynamics simulations have revealed that the presence of a stepped edge defect structure with an expanded layer spacing can enhance the transport of interfacial water molecules on the electrode surface by establishing sub-nanometer channels and tailoring a dynamic local micro-environment (Supplementary Fig. 11 and Supplementary Movie 2). This phenomenon contributes to the acceleration of charge transfer between active centers and water molecules as well as the migration of intermediate species, thereby enhancing the overall HER kinetics[53–56].

## Electrical and mechanical properties of WS$_2$ superstructure

It is well known that the electrical properties of catalysts are a key factor influencing electrocatalytic reactions, especially under industrial application conditions[57–59]. Atomic force microscopy (AFM) and Kelvin probe force microscopy (KPFM) were performed to investigate the electrical properties of the synthesized WS$_2$ superstructure. The AFM image presented in Fig. 3a clearly shows a single curved WS$_2$ superstructure formed by the orientation-dependent assembly of nanosheets. KPFM test reveals the surface potential of WS$_2$ superstructure on the Si substrate (Fig. 3b). As shown in Fig. 3c, the three different positions in the WS$_2$ superstructure (marked with different symbols) show a clear potential discrepancy, indicating the enrichment of surface electrons at the high strain concentration region. We also tested the surface potentials of individual WS$_2$ superstructure with different shapes in the same field of view (Supplementary Fig. 12). The result shows that the surface potential varies with the length and degree of curvature of the WS$_2$ superstructure. In addition, we investigated the electronic structure characteristics of WS$_2$ superstructure. Electron spin resonance (ESR) spectra of 2H-WS$_2$, 1T-WS$_2$, and WS$_2$ superstructure, as shown in Fig. 3d, were used to verify the discrepancy in electronic configuration. The increased ESR intensity further reveals the characteristics of significant unpaired electron enrichment in WS$_2$ superstructure crystals due to the electronic rearrangement between W-d and S-p orbitals. We further evaluated the vertical conductivity of WS$_2$ superstructure. Figure 3e shows a schematic diagram of a typical thin film device based on WS$_2$ superstructure using a polyethylene terephthalate (PET) substrate. We applied different voltages to the two ends of the device and recorded the instantaneous currents at room temperature. The conductivity of WS$_2$ superstructure is obtained using the conductivity calculation formula. For comparison, the room-temperature conductivities of 1T-WS$_2$/PET and pure PET substrates were also tested. As shown in Fig. 3f, the observed I-V curves for the two sample devices are linear and symmetrical, while the pure PET device shows non-linear and insulating properties (Supplementary Fig. 13). Interestingly, the room temperature conductivity of 11.8 S cm$^{-1}$ for WS$_2$ superstructure is much higher than that of 1T-WS$_2$ (2.9 S cm$^{-1}$), indicating its excellent conductivity[60]. This phenomenon may be attributed to the change in the energy band filling state and the Fermi energy level caused by intercalation, which significantly enhances the conductivity[61,62].

The mechanical properties of WS$_2$ superstructure were investigated using in situ SEM. We performed compression and extension on WS$_2$ superstructures with different shapes using nano-manipulator tip in SEM. Figure 3g, h demonstrates the evolution of the relative compression and extension operations with time, respectively. It can be clearly seen that the WS$_2$ superstructure was compressed into a flatter ellipse and subsequently returned to the initial state intact (Fig. 3g and Supplementary Movie 3). Similarly, the WS$_2$ superstructure was stretched and showed significant deformation as well as immediate recovery of strain after the withdrawal of the external force (Fig. 3h and Supplementary Movie 4). These results suggest that the assembled 2D nanosheets slide or rotate around each other through the bond-free

VDW interface, endowing the WS$_2$ superstructure with excellent mechanical flexibility and achieving sustainable absorption-release of external mechanical energy. Therefore, we can speculate that the WS$_2$ superstructure would undergo dynamic topological geometrical deformation in response to external mechanical energy (such as bubble forces and local electric field forces) involving high current density hydrogen production. This phenomenon not only facilitates efficient charge transfer but also maximizes the exposure of catalytically active sites during electrochemical reactions, thus contributing to efficient electrocatalytic hydrogen production in AEMWE.

## Electrochemical HER performance

We investigated the HER performance of WS$_2$ superstructure in hopes of meeting the demand for high-performance electrocatalysts in industrial-scale hydrogen production. The HER performance of WS$_2$ superstructure sample was evaluated using the conventional three-electrode system. Besides, the WO$_3$, 2H-WS$_2$, 1T-WS$_2$ nanosheets, and commercial Pt/C (20 wt%) were tested as comparison samples under the same conditions. The effect of carbon fiber cloth (CFC) support on current density in HER is negligible in this work (Supplementary Fig. 14a). It can be clearly seen from Fig. 4a and Supplementary Fig. 14b that WS$_2$ superstructure has the better HER performance compared with the other reference samples. Notably, the Pt/C exhibits superior performance compared to that of WS$_2$ superstructure at low current densities (<150 mA cm$^{-2}$) (Supplementary Fig. 15). As the current density increases, the WS$_2$ superstructure exhibits even better HER performance than Pt/C. Specifically, the overpotentials of the WS$_2$ superstructure electrode are as low as 205 mV at 500 mA cm$^{-2}$ and 264 mV at 1000 mA cm$^{-2}$, which represent a considerable reduction in contrast to that of Pt/C (269 mV@500 mA cm$^{-2}$ and 429 mV@1000 mA cm$^{-2}$), as shown in Fig. 4b. Note that the HER performance of Pt/C (20 wt%) used in this work is comparable to or better than that reported for Pt/C in the literature (Supplementary Table 1). To further evaluate the performance at high current density, we calculated the $\Delta\eta/\Delta\log|j|$ ratio, which was considered as a suitable indicator to evaluate the performance of the catalyst in the high current density range[11,12,14]. As shown in Fig. 4c, the $\Delta\eta/\Delta\log|j|$ value of Pt/C electrode grows significantly as the current density increases, with a value of 847 mV dec$^{-1}$ in the current density range of 1000–2000 mA cm$^{-2}$. In contrast, at the WS$_2$ superstructure electrode, the value is considerably lower at only 340 mV dec$^{-1}$@1000–2000 mA cm$^{-2}$, indicating its remarkable HER performance at high current density.

To better understand the excellent performance of WS$_2$ superstructure catalysts at high current density, we determined the electrode-electrolyte interfacial area (ECSA) using the double-layer capacitance (C$_{dl}$) method, and subsequently calculated the ECSA-normalized current density and turnover frequency (TOF)[63–67]. The ECSA value of WS$_2$ superstructure is 320.8 cm$^2_{ECSA}$, which is significantly larger than that of 1T-WS$_2$ (239.2 cm$^2_{ECSA}$) and 2H-WS$_2$ (13.7 cm$^2_{ECSA}$) (see Supplementary Note 1 and Supplementary Figs. 16–19 for detailed calculation and analysis). Surprisingly, the WS$_2$ superstructure catalyst exhibits impressively high values of TOF and ECSA-normalized current density in comparison to the reference samples and even surpasses the performance of most of the reported state-of-the-art HER electrocatalysts (Supplementary Figs. 20 and 21, Supplementary Tables 2–5, see Supplementary Note 2 for details). The electrochemical impedance spectroscopy (EIS) analysis was also performed to investigate the charge transfer mechanism of the HER reaction. The WS$_2$ superstructure sample exhibited remarkably lower charge transfer resistance (3.87 ± 0.03 Ω) than the other samples, 8.16 ± 0.04 Ω for 1T-WS$_2$ and 17.14 ± 0.04 Ω for 2H-WS$_2$, indicating its superior charge transfer kinetics (Fig. 4d, see Supplementary Table 6 for detailed impedance parameters). These results suggest that WS$_2$ superstructure catalyst possesses excellent intrinsic activity, abundant

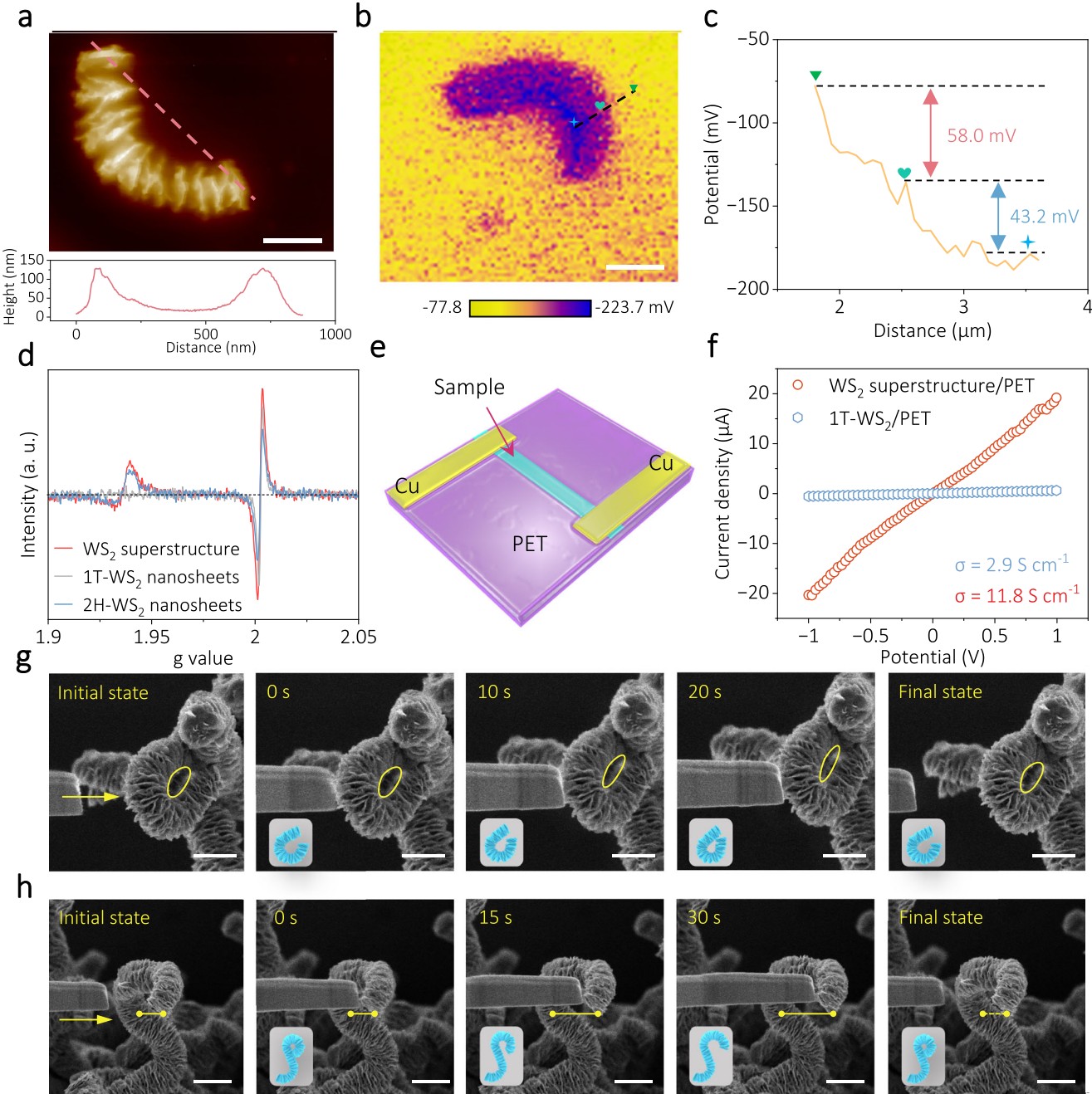

**Fig. 3 | Electrical and mechanical characterizations of WS₂ superstructure.**
**a** AFM image of single WS₂ superstructure. Bottom picture illustrates the height profile measured along the pink dashed line. Scale bar, 200 nm. **b, c** Surface potential of single WS₂ superstructure detected with KPFM. Scale bar, 1.0 μm. **d** ESR spectra of WS₂ superstructure, 1T-WS₂, and 2H-WS₂ samples. **e** A schematic illustration of WS₂ superstructure device. **f** Current-voltage curves for the device shown in (**e**). **g, h** In situ SEM mechanical test of WS₂ superstructure. Series of SEM images with the compression (**g**) and extension process (**h**) of a WS₂ superstructure. The inset shows the structure models of WS₂ superstructure at different stress states. Scale bar, 500 nm.

active sites, and superior interfacial charge transfer kinetics, which collectively contribute to its remarkable HER performance at high current density.

Evaluating the stability of electrocatalysts under high current density is crucial in determining their feasibility for industrial applications. The chronoamperometric measurements were carried out at different high current densities (500, 1000, and 2000 mA cm⁻²) to evaluate the HER stability of the WS₂ superstructure electrode. As displayed in Fig. 4e, the performance degradation of the electrode was

negligible for operation at different high current densities, suggesting its potential as a promising electrocatalyst for industrial applications that facilitate lossless transitions across multiple scenarios.

To understand the underlying factors contributing to the excellent activity of the WS₂ superstructure, in-situ surface-enhanced Raman spectroscopy (SERS) was performed to analyze surface reaction intermediates during the HER process[54]. As shown in Supplementary Fig. 22, the broad peak spanning from 3000 to 3700 cm⁻¹ corresponds to the adsorbed water peaks in the WS₂ superstructure

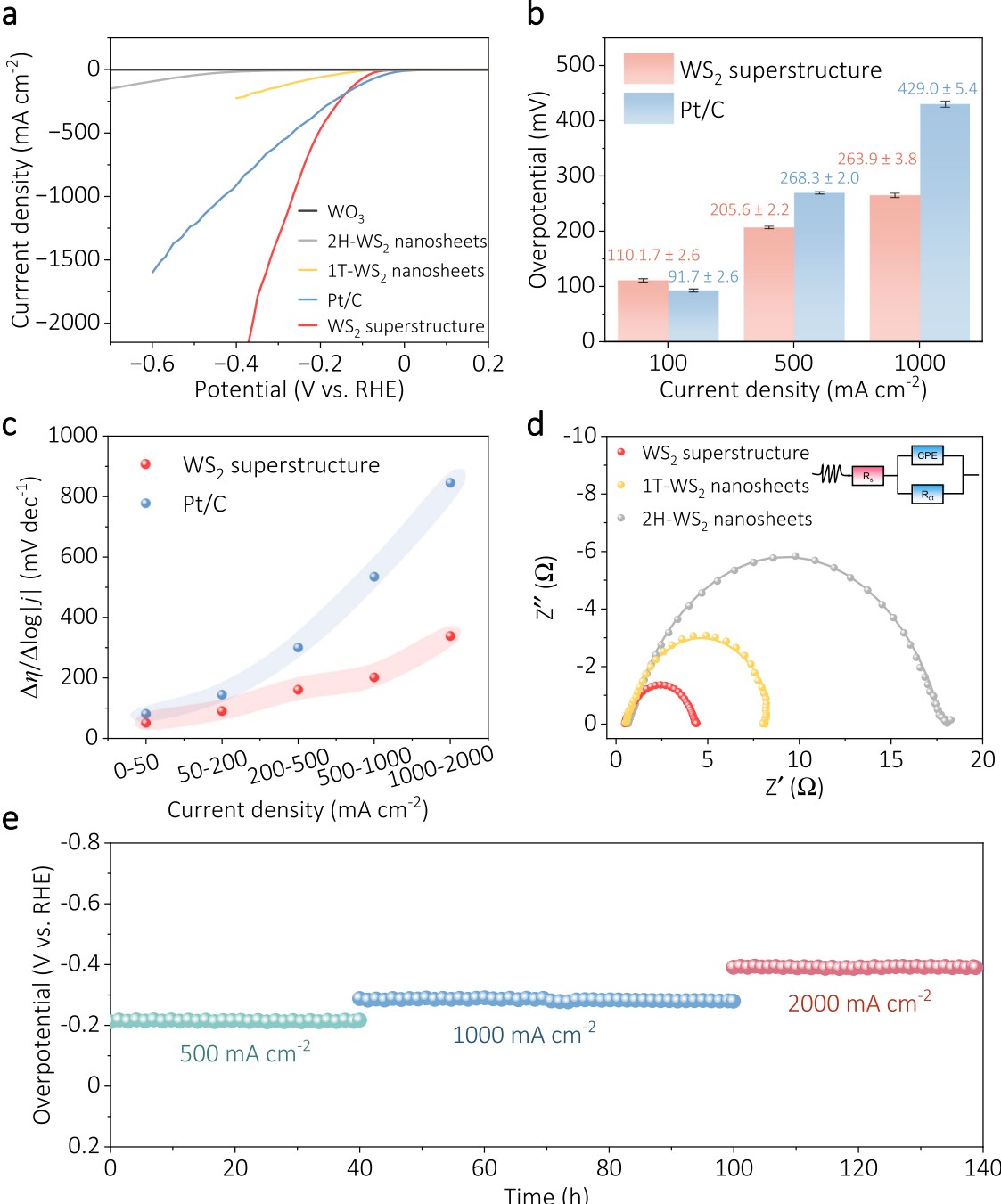

**Fig. 4 | High-current-density HER performance of WS$_2$ superstructure.**
**a** Polarization curves of different samples in 1.0 M KOH (scan rate: 10 mV s$^{-1}$, electrode surface area: 1 cm$^2$, after iR correction). **b** Histograms of overpotential at different current densities of WS$_2$ superstructure and Pt/C with error bar. Error bars corresponded to the standard deviation of three independent measurements. **c** $\Delta\eta$/$\Delta$log|$j$| ratios of the WS$_2$ superstructure, and Pt/C catalysts at different currents. **d** Nyquist plots of impedance data for various samples, using the frequency in the range from 100 kHz to 0.1 Hz at −50 mV (vs. RHE). Inset shows the equivalent circuit used for fitting the data. The fitting circuit of the corresponding EIS plot was used to determine charge transfer resistance ($R_{ct}$) (3.87 ± 0.03 Ω for WS$_2$ superstructure, 8.16 ± 0.04 Ω for 1T-WS$_2$, 17.14 ± 0.04 Ω for 2H-WS$_2$). **e** Chronoamperometric (i-t) curves of the WS$_2$ superstructure at various current densities.

sample, demonstrating its excellent water adsorption capacity[68]. This result is in good consistent with the DFT calculations (Fig. 2j). The successful observation of S-H bonds (-2550 cm$^{-1}$) via in-situ SERS directly confirmed that sulfur atoms serve as the catalytic active sites in WS$_2$ superstructure for HER[69–71]. In addition, the irreversible electrochemical oxidation study demonstrated the edge-rich characteristic of the WS$_2$ superstructure (Supplementary Fig. 23)[72]. The poison experiments using zinc nitrate further demonstrated that the edge-stepped

defects with abundant unsaturated sulfur atoms play an important role in determining the HER property of the WS$_2$ superstructure electrocatalyst (Supplementary Fig. 24)[69,73]. We can conclude from the in-situ SERS, irreversible electrochemical oxidation, and poison experiments that edge-stepped defects with abundant unsaturated sulfur atoms play an important role in determining the HER property of the WS$_2$ superstructure electrocatalyst. During the HER process, these stepped-like structures could provide abundant unsaturated sulfur atoms along

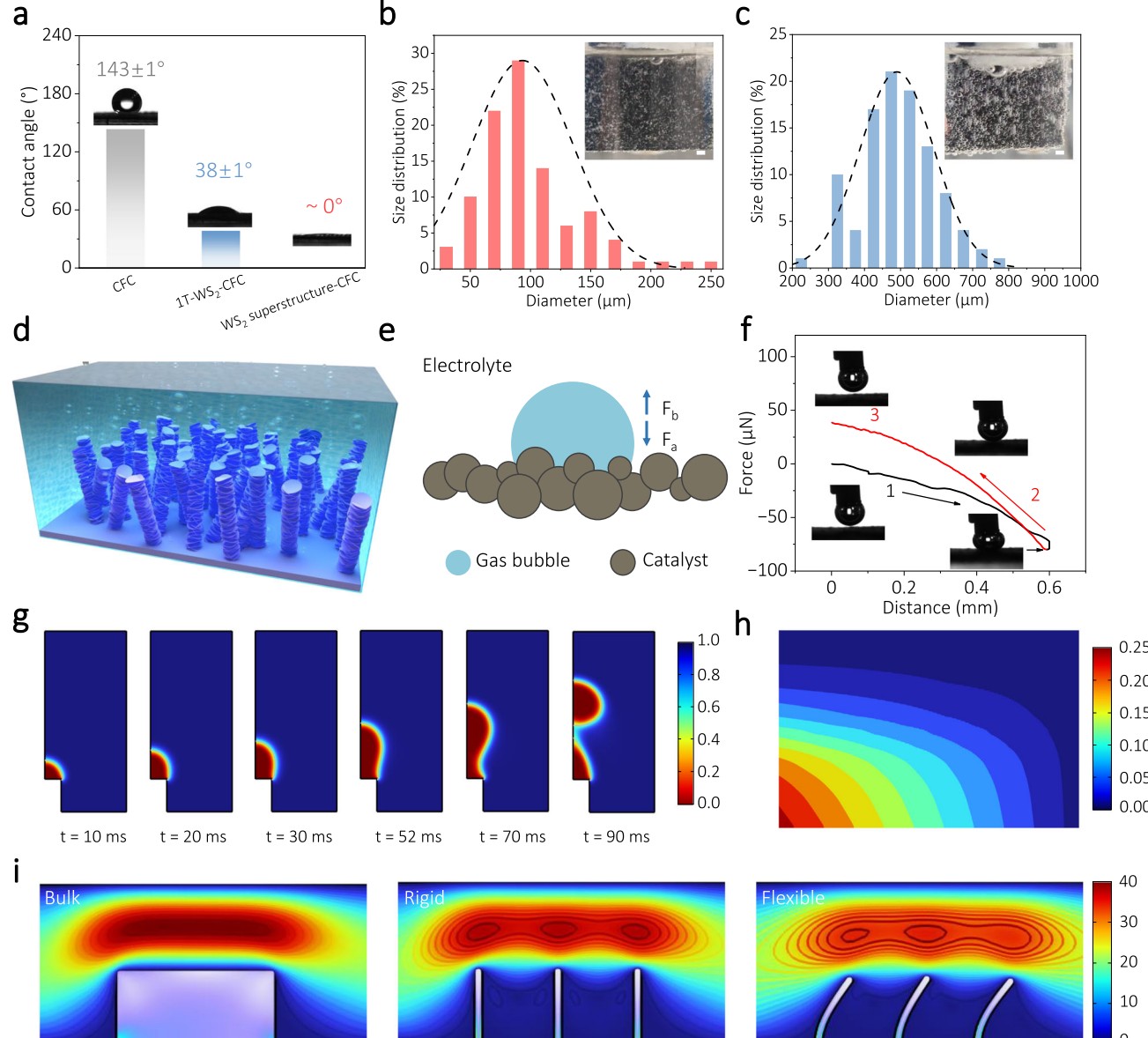

**Fig. 5 | Mass transfer behavior of electrocatalyst under high current density operation. a** Contact angles of an electrolyte droplet on the electrode surfaces. Size distribution statistics of released bubbles on the electrode surfaces, and (inert) photographs of the electrode during HER electrocatalysis (**b** WS$_2$ superstructure; **c** Pt/C). Scale bar, 1.0 mm. **d** Schematic illustration of H$_2$ bubble evolution on the WS$_2$ superstructure electrode surface. **e** Force analysis of gas bubble on the electrode surface. **f** Adhesive force measurement of the bubble on the WS$_2$ superstructure electrode surface. Insets display the bubble states during the testing process. **g** Hydrodynamic simulation of the complete bubble formation process on the hydrophilic surface. **h** Phase diagram of hydrogen volume fraction as a function of time in finite element simulations. **i** Multiphysics field simulation analysis based on fluid-structure interaction. The color bar shows the relative scale of the velocity field distribution.

with their curved edge profiles to serve as active centers to promote the evolution of H species.

**Mass transfer behavior**

Mass transfer behavior occurring at the phase interface, which includes the flow of liquid-phase reactants and the evolution of gas-phase products, is a crucial step of the electrochemical gas evolution reactions, particularly at high current density[11,59]. Combining experimental and theoretical models, we have systematically investigated the mass transfer behavior at the electrocatalyst interface. Contact angle (CA) measurements were carried out to evaluate the surface wettability of the as-prepared electrocatalysts[74]. As shown in Fig. 5a, the bare CFC substrate displays hydrophobic behavior with a CA of 143 ± 1°. In comparison, the 1T-WS$_2$@CFC and WS$_2$

superstructure@CFC surfaces exhibit hydrophilicity. Particularly striking is the superhydrophilicity of the WS$_2$ superstructure@CFC surface, where electrolyte droplets diffuse at an outstandingly rapid speed, reaching a near-zero contact angle within 0.5 s, as shown in Supplementary Movie 5. This phenomenon is attributed to the unique micro-nanostructure and metal 1T phase of WS$_2$ superstructure[60]. The highly effective and rapid diffusion ability resulting from the superhydrophilic of the WS$_2$ superstructure@CFC surface is key for accelerating the flow of electrolytes, contributing to the achievement of industrial-level high current density.

The gas evolution on the electrode surface is considered to be a crucial step in the HER process at high current density[75,76]. To observe this process, the high-speed camera was used to record the bubble evolution on the surface of Pt/C@CFC and WS$_2$ superstructure@CFC

electrodes. Figure 5b shows that the bubbles generated on the surface of WS$_2$ superstructure@CFC electrode usually measure less than 200 μm in diameter and depart rapidly from the electrode surface upon formation (Supplementary Movie 6). In comparison, the bubbles on the surface of Pt/C@CFC electrode often reach diameters greater than 400 μm before separating from the electrode surface (Fig. 5c). However, the slow growth and detachment of these relatively large bubbles can create a "dead zone" in the electrocatalyst active sites, which blocks contact with the electrolyte solution for an extended period and leads to the deterioration of the HER performance (Supplementary Fig. 25)[74,77]. It is proposed that the dynamic variable edge microstructure of the WS$_2$ superstructure@CFC electrode exerts a cutting effect on the gas-liquid-solid interface during HER process, displaying a dynamically adaptive characteristic upon bubbling[78,79] (Fig. 5d, Supplementary Figs. 26 and 27, and Supplementary Movie 7), ultimately leading to smaller bubble sizes and reduced adhesion forces. To further verify this conclusion, the adhesion force measurements of a single bubble to the electrode surface in the electrolyte medium were conducted. During the evolution process, the bubble is subjected to both the adhesion force ($F_a$) and the buoyancy force ($F_b$) of the electrode surface, as shown in Fig. 5e. As the size of the bubble grows, the buoyancy force gradually increases and overcomes the adhesion force, leading to a detachment of the bubble from the electrode surface. Upon departure of the bubble from the 1T-WS$_2$@CFC electrode surface, there is a clear indication of a "cliff" in the bubble's force curve ($F_b = 20.9$ μN) with a visible deformation of the bubble (Supplementary Fig. 28). Conversely, the bubble force curve shows a negligible bubble adhesion ($F_b \approx 0$ μN) on the surface of WS$_2$ superstructure@CFC with barely detectable deformation of bubble in the inset (Fig. 5f). Therefore, during HER under high current density, H$_2$ bubbles can detach from the electrode surface easily with minimal adhesion force, maintaining maximal achievable active site reactivation efficiency, thereby providing a stable and pronounced increase in HER current.

To reveal the corresponding relation between electrode surface hydrophilicity and H$_2$ bubble evolution, finite element simulations were performed to investigate the growth and desorption processes of interfacial bubbles using a separated multiphase flow model described by the interface tracking method. The complete simulation system comprises three essential components: the hydrophilic electrode interface, the electrolyte layer, and the bubble topological phase. It is noteworthy that the bubble evolution process is simulated utilizing a moving boundary that changes topologically with time. In the whole simulation system, the transport of mass and momentum is controlled by the incompressible Navier-Stokes equations (more details are seen in Supplementary Note 3). The evolution of the hydrogen bubble interface topology can be seen in Fig. 5g, and it is clear that the growth and detachment of individual bubble from the substrate occur at around 90 ms. Compared with hydrophobic electrode surfaces, the higher hydrophilicity of the electrode surface (i.e., lower CAs) is found to accelerate the rate of bubble evolution on the electrode surface (Supplementary Fig. 29). Moreover, Fig. 5h and Supplementary Fig. 30 show the evolution of the gas volume fraction with time, indicating that the kinetic and mass balance in the simulated system remain homogeneous over time.

We build the fluid-structure interaction models to investigate the corresponding relation between electrode flexibility and mass transfer efficiency (more details are seen in Supplementary Note 4). The fluid-structure interaction between the electrode materials with different characteristics and the electrolyte is presented in Fig. 5i. Compared with the bulk or the rigid-type electrodes, the flexible electrode induced three weaker vortices, indicating the greater velocity field variation in the whole simulation system. The simulation results revealed that the dynamic deformation of flexible electrode greatly disturbed the electrochemical reaction region, bringing about

intensified convection of the reaction components and thus enhancing the mass transfer efficiency at the multiphase interface. These results clearly demonstrate that the constructed WS$_2$ superstructure@CFC electrode is essential to promote the mass transfer (liquid reactants and H$_2$ bubbles) behavior at the interface, leading to remarkable and stable HER performance at high current density.

## Performance of AEM water electrolyzer devices

To test the viability of WS$_2$ superstructure for large-scale hydrogen production, a flow-type AEM electrolyzer was assembled using WS$_2$ superstructure as the cathode and the commercial IrO$_2$ as anode (Fig. 6a, b). The steady-state polarization curve shows the excellent performance of the AEM electrolyzer catalyzed by WS$_2$ superstructure in comparison with that catalyzed by commercial Pt/C catalyst and conventional WS$_2$ nanosheets (Fig. 6c and Supplementary Fig. 31). Specifically, the AEM electrolyzer (WS$_2$ superstructure || IrO$_2$) needs a cell voltage of only 1.701 V to achieve a current density of 1 A cm$^{-2}$, far superior to that obtained with commercial Pt/C || IrO$_2$ (1.757 V@1 A cm$^{-2}$) and conventional WS$_2$ nanosheets || IrO$_2$ (1.934 V@1 A cm$^{-2}$). The long-term operational stability is shown in Fig. 6d. Impressively, the WS$_2$ superstructure-based cell demonstrates well-maintained AEM electrolyzer performance after running under 1 A cm$^{-2}$ for 1000 h at 60 °C, with slight increase (ca. 9.67 μV h$^{-1}$, meaning a cell voltage increase as low as 235.6 mV over one year working time) in cell voltage, far better than conventional WS$_2$ nanosheets (Supplementary Fig. 31d). Such remarkable stability not only sets a new benchmark for non-precious metal-based catalysts but also surpasses that of some advanced platinum group metal cathode catalysts recently reported (Supplementary Table 7). In addition, we systematically characterized post-catalysis WS$_2$ superstructure by SEM, TEM, in-situ SEM, XRD, and XPS spectroscopy. The morphology and structure of this cathode catalyst after continuous operation for 1000 h were found to be well retained in the AEM electrolyzer (Supplementary Figs. 32–37).

Given that reaction parameters such as pressure and temperature have a significant influence on the actual reaction process in a practical multiphase electrochemical reaction system[80,81], we constructed a finite element model to analyze the operating state of the electrode material at a practical water electrolyzer. The mass transport of hydrogen and water within the cathode compartment of the electrolytic cell model is primarily based on the mechanisms of Maxwell-Stefan diffusion and the Brinkman equation. The simulation results show that the hydrogen gas generated undergoes inhomogeneous diffusion at the cathode reaction side, and the local partial pressure of the hydrogen gas exhibits a gradient-type distribution, resulting in a periodic pressure effect on the electrode material during the water electrolysis process (Fig. 6e). The recurring fluctuation of hydrogen pressure within the cathode chamber has the potential to cause mechanical damage to the electrode material through abrasion inside the electrolyzer[82,83]. Besides, after prolonged operation, the electrode material generates thermal strain due to the Joule heating effect[84,85], which causes significant macroscopic deformation and may lead to mechanical losses of the material (Fig. 6f, g and Supplementary Fig. 38). As a result, designing the WS$_2$ superstructure with excellent mechanical flexibility is expected to effectively accommodate mechanical losses caused by volumetric expansions and contractions at prolonged industrial high current density, enabling stable hydrogen production. It is concluded, therefore, that the WS$_2$ superstructure catalyst holds the highlighted practical application prospects for industrial hydrogen production.

## Discussion

In summary, we have shown that the designed WS$_2$ superstructure can be used as HER catalysts that show unparalleled alkaline HER performance at ampere-level current density hydrogen production.

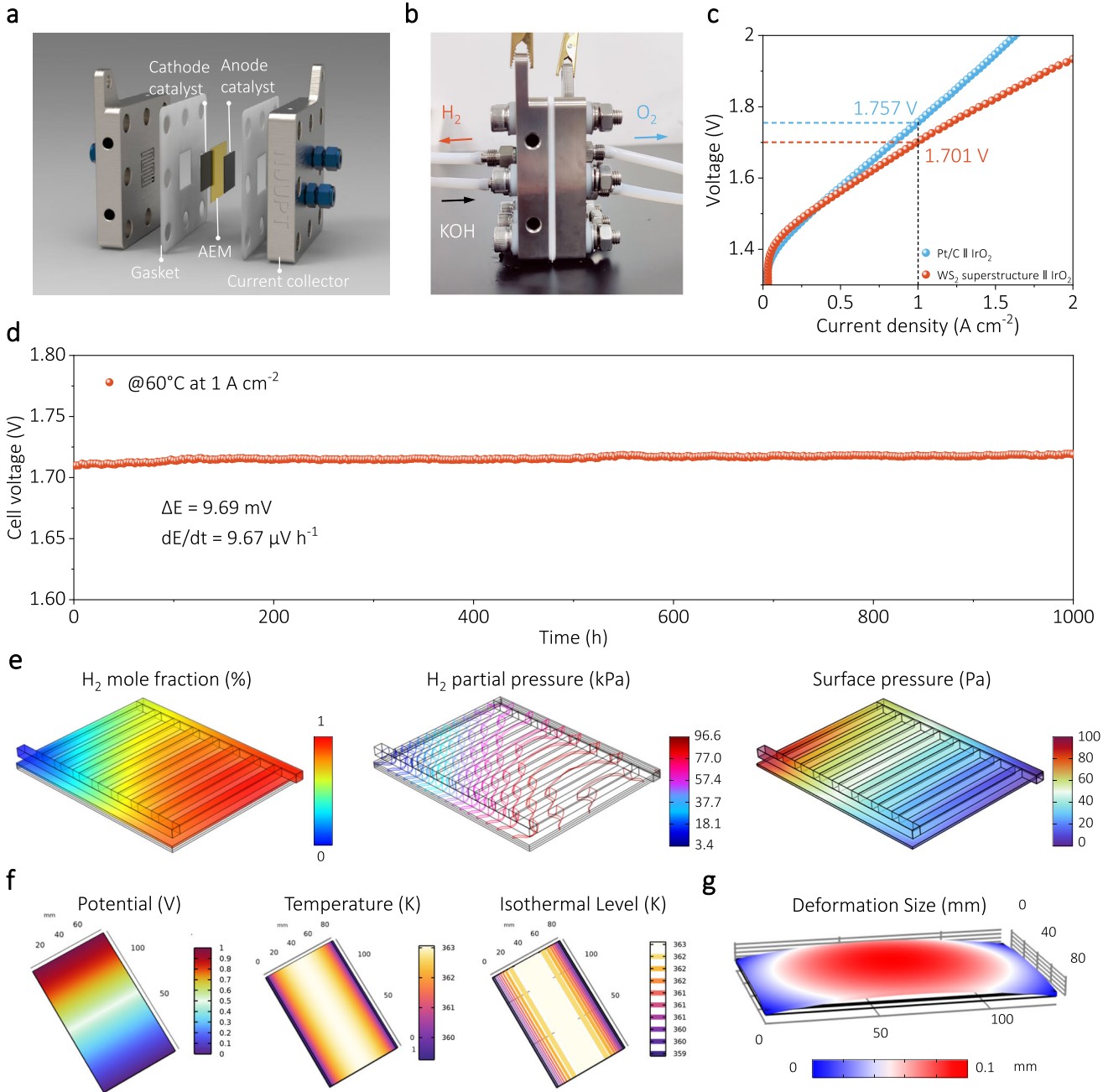

**Fig. 6 | AEM electrolyzer performance using WS₂ superstructure as catalyst. a, b** Schematic diagram and optical photo of the AEM electrolyzer. **c** Polarization curves of the AEM electrolyzer. **d** Chronopotentiometric curve of the AEM electrolyzer using WS₂ superstructure catalyst. **e** Computational fluid dynamics simulation of cathode side of electrolysis cell model. Phase-field simulations of Joule heating (**f**) and strain response (**g**) distribution in the cathode element of electrolysis cell.

Remarkably, the AEM electrolyzer using WS₂ superstructure cathode achieves a low cell voltage of 1.70 V at 1 A cm⁻² under industrial testing (30 wt% KOH at 60 °C), while maintaining stable electrolysis for 1000 h at 1 A cm⁻² with performance degradation rate only at 9.67 μV h⁻¹. In-situ SEM mechanical tests, finite element simulations and other characterization techniques reveal that the WS₂ superstructure has extraordinary mechanical properties and excellent electrical conductivity, as well as hydrophilic/aerophobic surface properties, which substantially enhance the catalytic activity and mass transfer kinetics involving high current density and continuous dense gas evolution. The design concept demonstrated in this work highlights the significance of designing electrocatalysts for operation at high current density as required by industry and further promotes their application prospects in industrial-relevant electrolytic cells.

## Methods

### Chemicals
All chemicals are of analytical grade and are used without further treatment. Tungsten (VI) chloride, sulfur power and Nafion (5 wt%) was purchased from Aladdin Biochemical Technology Co., Ltd. Octylamine and ethanol were purchased from Sinopharm Chemical Reagent Co., Ltd. Commercial IrO₂, Pt/C (40 wt%), anion exchange membrane (Sustainion X37-50 Grade 60) and carbon fiber cloth (CFC) were obtained from SCI Materials Hub.

## Preparation of WS$_2$ superstructure

In a typical experiment, 0.4 mmol WCl$_6$·xH$_2$O, 3.6 mmol sulfur power, 18 mL ethanol, and 10 mL octylamine were mixed thoroughly in 100 mL Teflon-lined autoclave. The mixed solution was warmed to 200 °C and held for 48 h, and then the whole system was gradually cooled to ambient temperature. Finally, the resulting black samples were centrifuged, washed (ethanol), and vacuum dried (60 °C) to obtain the WS$_2$ superstructure sample. WO$_3$ nanorods were synthesized in the same ratio of mixed solvents, except that the reaction time was changed to 24 h.

## Preparation of 1T-WS$_2$ and 2H-WS$_2$

1T-WS$_2$ sample (MK407) and 2H-WS$_2$ were purchased from Nanjing MKNANO Tech. Co., Ltd as the reference sample.

## Materials characterization

SEM and TEM images were measured using a Hitachi S-4800 and Hitachi HT7700. More detailed morphology and atomic structure were characterized by FEI Talos F200X (acceleration voltage of 200 kV). XRD dates were recorded by using Bruker AXS D8 Advance A25 ($2\theta = 5.0–80.0°$). XPS test was conducted using Thermo Fisher Scientific K-Alpha with an X-ray photoelectron spectrometer. Raman spectra were carried out on a confocal Raman microscope (inVia, Renishaw, England). AFM characterizations were performed from a commercial Bruker Dimension Icon with KPFM mode. Static contact angle and bubble adhesion force of samples were measured by the KRUSS (DSA20) and Dataphysics DCAT25 system. In-situ SEM test on WS$_2$ superstructure was performed using FIB (Helios NanoLab DualBeam) with Kleindiek mechanical manipulators.

## Electrochemical characterization in a three-electrode system

The electrochemical performance was measured using a conventional three-electrode system on a CS310 workstation (Wuhan Corrtest Instrument Corp., Ltd). Data were collected and processed using the computer software CS Studio 5 and Origin, respectively. Hg/HgO electrode and a graphite rod were used as the reference and counter electrodes to evaluate the electrochemical performance of all samples. The H$_2$-bubbled 1.0 M KOH solution were used as the electrolytes during electrochemical measurement. To ensure the accuracy of the electrochemical tests, the electrolyte used in this paper was prepared immediately prior to use, and the pH of the electrolyte was measured several times before the test. The potential was referred to reversible hydrogen electrode (RHE) by calibrating the Hg/HgO electrode with Pt foil as both the working and the counter electrodes in a sealed standard three-electrode. Cyclic voltammograms (CV) were tested at the scan rate of 1 mV s$^{-1}$ and the average of the two inter-conversion point values was taken to be the thermodynamic potential for HER (-0.916 V). Practical pH value of 1 M KOH in this work was measured to be 13.84 ± 0.12. When the measured pH value is adopted for the calculation using the equation $E_{RHE} = E_{Hg/HgO} + 0.098\,V + 0.059 \times pH$, the calculated calibrated potential is determined to be 0.914 ± 0.007 V, which is close to the measured value (0.916 V). The presented potentials were calibrated to the RHE using the measured RHE-calibrated potential value. Catalyst dispersion was fabricated by mixing catalytic sample (5.0 mg), Nafion (20 μL, 5 wt%), deionized water (800 μL), and absolute ethanol (200 μL) thoroughly. Then, 40 μL of the dispersion was dropped onto the carbon fiber cloth (CFC) (1 cm$^2$) and dried for about 12 h. Specifically, the mass loading of the WS$_2$ superstructure electrocatalyst is -0.196 mg cm$^{-2}$. Polarization curves were measured in the potential range of −0.75 – 0.20 V vs. RHE with a sufficiently slow sweep rate. All polarization curves were corrected by 100% iR compensation based on measured electrolyte resistance. The Tafel slope was obtained by plotting the overpotential as a function of logarithm of current. Cyclic voltammetry (CV) was used to test the electrochemical active surface areas (ECSA) of catalysts. Detailed analysis and

calculation were shown in Supplementary Note 1. The chronoamperometry method was employed to study the electrochemical stability of catalyst. Electrochemical impedance spectroscopy (EIS) test was conducted from 100 kHz to 0.1 Hz at −50 mV vs. RHE, and the alternating current amplitude was 5 mV. The fitting circuit of the corresponding EIS plot (Fig. 4d) was used to determine the $R_{ct}$ (3.87 ± 0.03 Ω for WS$_2$ superstructure, 8.16 ± 0.04 Ω for 1T-WS$_2$, 17.14 ± 0.04 Ω for 2H-WS$_2$) and $R_s$ (0.48 ± 0.05 Ω for WS$_2$ superstructure, 0.56 ± 0.08 Ω for 1T-WS$_2$, 0.69 ± 0.10 Ω for 2H-WS$_2$).

## AEM measurement

The AEM electrolyzer was mainly composed of an anode (4.0 cm$^2$), a cathode (4.0 cm$^2$), gas diffusion layer (carbon paper) and AEM (Sustainion X37-50 Grade 60). Membrane thickness and size are 50 μm and 6.25 cm$^2$, respectively. Note that the membrane has been activated for 24 h in 1 M KOH before use. To fabricate the membrane electrode assembly (MEA), WS$_2$ superstructure, conventional WS$_2$ nanosheets or commercial Pt/C (40 wt%) were used as the cathode catalysts, and commercial IrO$_2$ as the anode catalyst. The catalyst was dispersed in a mixture of isopropanol and deionized water (4:1) and ultrasonicated for 10 min. Subsequently, Nafion solution (5 wt%) was added to the mixture and ultrasonicated for 2 h to obtain a homogeneous catalyst ink. This ink was subsequently applied to both sides of the AEM through ultrasonic spray technique. The assembled MEAs were sandwiched in a polytetrafluoroethylene spacer and subjected to hot pressing at 130 °C and 2 MPa for 3 min, followed by an additional 30 s pressing after cooling to room temperature. A Ti-fiber felt was used as the anodic gas diffusion layer (GDL), and carbon paper as the cathodic GDL. We investigated the water electrolysis performance of the AEM electrolyzer in 30 wt% KOH electrolyte at 60 °C using an electrochemical workstation (CS310, Wuhan Corrtest Instrument Corp., Ltd). The durability test of the AEM electrolyzer was performed under 1 A cm$^{-2}$ for 1001 h at 60 °C.

## Data availability

The data that support the findings of this study are available from the corresponding authors upon reasonable request.

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

## Acknowledgements

This work was financially supported by the National Funds for Distinguished Young Scientists (61825503) (Q.Z.), Postgraduate Research & Practice Innovation Program of Jiangsu Province (KYCX22_1009) (L.B.X.). Innovation Support Programme (Soft Science Research) Project Achievements of Jiangsu Province (BK20231514) (Q.Z.). The authors acknowledge the help and support of Prof. J.T. and Mr. X.L.H. from Fuzhou University for conducting in situ surface-enhanced Raman tests.

## Author contributions

Q.Z. and L.L.W. conceived the project. L.L.W. designed, supervised and analyzed the project. L.B.X. carried out the materials synthesis, characterizations, electrochemical tests, theoretical simulations, and all related data analysis. J.M.C. carried out the Raman characterization and data analysis. X.L. carried out the theoretical calculation and molecular dynamics simulation. L.L.W. and L.B.X. wrote the manuscript. Q.Z., L.L.W., L.B.X., J.M.C., X.X.W., W.W.Z., and S.J.L. discussed and revised the manuscript. All the authors contributed to the discussion during the project.

## Competing interests

The authors declare no competing interests.
