## [Peer Review File · Nature Communications]

REVIEWER COMMENTS

Reviewer #1 (Remarks to the Author):

Zhao and co-workers have developed WS₂ superstructures and deployed them as cathodic material in AEM electrolyzers. The WS₂-based cathode displayed excellent HER activity, even superseding Pt/C at higher current density. Hence, this article showcased a practical solution for AEM electrolyzers. However, the authors should comment on the following before this article can be considered further: WS₂ has been widely used in literature for cathodic reaction. This background story is missing in the introduction portion. Hence, the evolution of earlier 2D- and 3D-WS₂ to this current superstructure is not clear.

The authors have extensively studied the physical interaction and properties of the WS₂ superstructure. However, the authors should also probe the structural changes in the WS₂ superstructure post-electrolysis in the electrolyzer.

The authors should perform a techno-economic analysis for the catalyst and provide a comparative picture with the current state-of-the-art electrocatalysts.

Reviewer #2 (Remarks to the Author):

Electrocatalytic hydrogen evolution reaction (HER) is an important reaction for energy storage/conversion. Actually, it is a bottleneck to produce hydrogen energy economy through water splitting. Therefore, an efficient HER electrocatalyst is critical for development of hydrogen electrochemistry. In this work, author reported WS₂ superstructure with highly catalytic property of hydrogen evolution in alkaline solution. The as-synthesized WS₂ superstructure can produce industrial-level current density for hydrogen generation. Remarkably, the AEM electrolyser using WS₂ superstructure cathode achieves an ultralow cell voltage of 1.70 V at 1 A cm⁻² under industrial testing (30 wt% KOH at 60°C). However, similar works on WS₂ HER electrocatalysts have been published, such as *Nat. Mater.*, 2013,12, 850–855; *ACS Nano* 2019, 13,10448–10455; *Energy Environ. Sci.*, 2014,7, 2608-2613; *Nat. Commun.*, 2021, 12, 709; *Small*, 2020, 16, 1905000, and so on. Therefore, this work lacks enough novelty, and the manuscript does not bring a compelling scientific case that connects structural characterizations and electrochemical findings. Therefore, the manuscript doesn't meet the high standard of Nature Communication in its present state. The current submission can be further improved and published elsewhere if the authors can address the following issue.

1. This paper seems to list only the electrocatalytic properties of WS₂ catalysts in its present state. Based on geometric area normalized LSVs, WS₂ catalysts exhibit a superior HER performance relative to Pt/C. However, ECSA-normalized current density and turnover frequency (TOF) of WS₂ and Pt/C is also compared.
2. The determination of active centers should be further discussed based on more experiments and DFT calculations. In-situ XRD and Raman measurements should be carried out to confirm the real active sites for HER.
3. Beside from the kinetics, author should explain why WS₂ catalysts show an extraordinary HER property compared to other WS₂ electrocatalysts reported.

4. The authors stated edge-stepped defects plays an important role in determining the HER property of WS₂ electrocatalyst. However, there is no experiment results to confirm it. How to control the edge-stepped defects in WS₂ electrocatalyst.

5 On the one hand, the authors think edge-stepped defects can improve the WS₂ electrocatalyst. On the other hand, the authors explored the effect of mass transfer behavior of WS₂ electrocatalyst on promoting the HER property.

Reviewer #3 (Remarks to the Author):

This manuscript reports the development of flexible WS₂ superstructure catalyst and determined the HER activity. The electrocatalytic performances are really impressive. In addition, the authors have performed an AEM water electrolyzer with impressive cell voltage and stability. However, there are some shortcomings in the work. The work can be considered after the addressing the following queries, which the authors may find is useful.

1. The authors are recommended to add more discussion about the 1T/2H phase of WS₂ and how they are structurally different from the superstructure.
2. What is the role of octylamine in the synthesis of WS₂ superstructure and what is the reason to specifically choose octylamine instead of simple amines? Do these amines play any roles in the morphology of the material.
3. The PXRD of the as-synthesized material is bit noisy, it is recommended to perform again. In addition, the PXRD patterns of individual 1T and 2H phases also need to be compared with the superstructure.
4. What is the reason behind the shifting of 002 planes at lower angles? Is there intercalation of any foreign moiety? Authors may try FTIR study to confirm that.
5. What is the reason for caterpillar-like morphology displaying better “mechanical flexibility” compared to other available morphologies.
6. It is highly recommended to mention and include more discussion in the main text about the developed phases (1T/2H) of WS₂ superstructure from the XPS spectrum. Further, please mention how superstructure is different from existing 1T/2H phase
7. The author should add more discussion on how the defect structure of WS₂ improves the overall activity of the WS₂ superstructure.
8. Please mention how the Tafel slope is estimated. If it is calculated from dynamic LSV, then the author should consider the backward LSV with 100% iR correction.
9. The authors have used Ag/AgCl as a reference electrode, which is highly unstable during long-term stability tests. Therefore, it is recommended to perform a stability study using Hg/HgO as a reference electrode.
10. Please include the fitting circuit and the corresponding fitting parameters of the corresponding EIS plot in Figure 4d. In addition, also mention the potential at which the EIS was performed.
11. The authors mentioned that the superstructure exhibits the “Cutting Edge effect”; could the author clarify this in detail.
12. The author should perform the high-resolution post-XPS study of the WS₂ superstructure to investigate any possible alteration in the chemical state of the W and S. Also determine the relative ratio

of 1T and 2H in the XPS spectra after prolonged stability.

Point-by-point response to the reviewers' comments

We sincerely thank the reviewers for carefully reviewing our manuscript and their valuable comments, which certainly help us improve our manuscript. We also thank the editor for giving us such an opportunity to address the comments and revise the manuscript. The changes in the revised manuscript have been highlighted in red for your review. The point-by-point responses are presented below, and our responses are in blue text.

Reply to Reviewer 1 and revisions made accordingly:

Zhao and co-workers have developed WS₂ superstructures and deployed them as cathodic material in AEM electrolyzers. The WS₂-based cathode displayed excellent HER activity, even superseding Pt/C at higher current density. Hence, this article showcased a practical solution for AEM electrolyzers. However, the authors should comment on the following before this article can be considered further:

Response: Thanks for your positive evaluation of this work and valuable suggestion. We have made revisions according to each comment, as summarized below.

1. WS₂ has been widely used in literature for cathodic reaction. This background story is missing in the introduction portion. Hence, the evolution of earlier 2D- and 3D-WS₂ to this current superstructure is not clear.

Response: Thanks for the comments and suggestions. We have added the background story in the introduction portion to clarify the evolution of earlier 2D-WS₂ to this current 3D superstructure (page 4, line 69–85). For your convenience, the added texts in the introduction portion are as follows:

“Tungsten disulfide (WS₂), as a typical representative two-dimensional (2D) TMDs, is widely regarded as a promising non-precious metal HER catalyst due to its high

intrinsic activity at edge sites^{22–24}. However, despite extensive efforts in recent years, its high HER activity remains confined to acidic media, with kinetics becoming quite sluggish in more practically viable alkaline environments^{22–27}. In addition, during the HER process, 2D-WS₂ nanostructures tend to form aggregated materials due to van der Waals (VDW) forces, leading to incomplete exposure of edge sites and consequently diminishing catalytic performance^{28,29}. More importantly, 2D-WS₂ nanostructures face great challenges in withstanding the mechanical abrasion induced by high-density gas-liquid exchanges, particularly at high current densities (amperage levels), to meet the demands of practical AEM electrolyser applications^{8,17,18}. Some studies have found that designing three-dimensional (3D) WS₂ structures could significantly maximize the exposure of active edges, improve reactant accessibility, and enhance mechanical stability, which is an effective strategy for achieving high-level HER performance^{30–33}. Nevertheless, until now, the development of WS₂-based cathodic electrocatalysts suitable for efficient and long-term hydrogen production in practical AEM electrolyser has not yet been realized.”

- [8] Chen, P., Hu, X. High-efficiency anion exchange membrane water electrolysis employing non-noble metal catalysts. *Adv. Energy Mater.* **10**, 2002285 (2020).
- [17] Yu, X. et al. “Superaerophobic” nickel phosphide nanoarray catalyst for efficient hydrogen evolution at ultrahigh current densities. *J. Am. Chem. Soc.* **141**, 7537-7543 (2019).
- [18] Zhai, P. et al. Engineering active sites on hierarchical transition bimetal oxides/sulfides heterostructure array enabling robust overall water splitting. *Nat. Commun.* **11**, 5462 (2020).
- [24] Lukowski, M.A., Daniel, A.S., English, C.R., Meng, F., Forticaux, A., Hamers, R.J., Jin, S. Highly active hydrogen evolution catalysis from metallic WS₂ nanosheets. *Energy Environ. Sci.* **7**, 2608-2613 (2014).
- [25] Sarma, P.V., Kayal, A., Sharma, C.H., Thalakulam, M., Mitra, J., Shaijumon, M.M. Electrocatalysis on edge-rich spiral WS₂ for hydrogen evolution. *ACS Nano* **13**, 10448-10455 (2019).

- [26] Kim, H.U., Kanade, V., Kim, M., Kim, K.S., An, B.S., Seok, H., Yoo, H., Chaney, L.E., Kim, S.I., Yang, C.W., Yeom, G.Y., Whang, D., Lee, J.H., Kim, T. Wafer-scale and low-temperature growth of 1T-WS₂ film for efficient and stable hydrogen evolution reaction. *Small* **16**, e1905000 (2020).
- [27] Han, A., Zhou, X., Wang, X., Liu, S., Xiong, Q., Zhang, Q., Gu, L., Zhuang, Z., Zhang, W., Li, F., Wang, D., Li, L.J., Li, Y. One-step synthesis of single-site vanadium substitution in 1T-WS₂ monolayers for enhanced hydrogen evolution catalysis. *Nat. Commun.* **12**, 709 (2021).
- [28] Tan, C., Luo, Z., Chaturvedi, A., Cai, Y., Du, Y., Gong, Y., Huang, Y., Lai, Z., Zhang, X., Zheng, L., Qi, X., Goh, M.H., Wang, J., Han, S., Wu, X.J., Gu, L., Kloc, C., Zhang, H. Preparation of high-percentage 1T-phase transition metal dichalcogenide nanodots for electrochemical hydrogen evolution. *Adv. Mater.* **30**, 1705509 (2018).
- [29] Han, W., Liu, Z., Pan, Y., Guo, G., Zou, J., Xia, Y., Peng, Z., Li, W., Dong, A. Designing champion nanostructures of tungsten dichalcogenides for electrocatalytic hydrogen evolution. *Adv. Mater.* **32**, e2002584 (2020).
- [30] Hu, X., Liu, Y., Li, J., Wang, G., Chen, J., Zhong, G., Zhan, H., Wen, Z. Self-assembling of conductive interlayer-expanded WS₂ nanosheets into 3D hollow hierarchical microflower bud hybrids for fast and stable sodium storage. *Adv. Funct. Mater.* **30**, 1907677 (2019).
- [31] Zeng, J., Zhang, L., Zhou, Q., Liao, L., Qi, Y., Zhou, H., Li, D., Cai, F., Wang, H., Tang, D., Yu, F. Boosting alkaline hydrogen and oxygen evolution kinetic process of tungsten disulfide-based heterostructures by multi-site engineering. *Small* **18**, e2104624 (2022).
- [32] Ji, L., Cao, H., Xing, W., Liu, S., Deng, Q., Shen, S. Facilitating electrocatalytic hydrogen evolution via multifunctional tungsten@tungsten disulfide core-shell nanospheres. *J. Mater. Chem. A* **9**, 9272-9280 (2021).
- [33] Xie, L., Wang, L., Liu, X., Zhao, W., Liu, S., Huang, X., Zhao, Q. Tetra-coordinated W₂S₃ for efficient dual-pH hydrogen production. *Angew. Chem. Int.*

Ed. **63**, e202316306 (2024).

2. The authors have extensively studied the physical interaction and properties of the WS₂ superstructure. However, the authors should also probe the structural changes in the WS₂ superstructure post-electrolysis in the electrolyzer.

Response: Thank you very much for the comments and suggestions. We agree with you that the post-electrolysis analysis in the electrolyzer of the WS₂ superstructure is very necessary. In this revision, we have supplemented more physical characterizations after the stability test, along with more discussions to probe the structure changes.

We newly collected XPS spectra (**Figure R1**), ICP-OES analysis (**Table R1**), and in-situ SEM images (**Figure R2**) for the WS₂ superstructure after the stability test in anion exchange membrane (AEM) electrolyzer.

As for the W 4f spectra (**Figure R1a**), the signals corresponding to the W⁴⁺ valence states in WS₂ superstructure showed no obvious shifts in binding energies after HER operation in the AEM electrolyzer. This result indicates that the W valence state in WS₂ superstructure was relatively stable after AEM electrolyzer operation. Similarly, the S 2p XPS signals in **Figure R1b** demonstrate that the S chemical state in WS₂ superstructure was relatively stable after AEM electrolyzer operation. Furthermore, we monitored W dissolution from the WS₂ superstructure catalyst by inductively coupled plasma optical emission spectroscopy (ICP-OES) after the stability test in AEM electrolyzer (**Table R1**). From the comparison in pre- and post-electrolysis electrolyte analysis (detection limit = 0.1 ppm), a slight increase in W concentration was observed in the solution after the stability test, which could be ascribed to the dissolution of the oxidized W species on the WS₂ superstructure surface (*Nat. Commun.* **2020**, *11*, 4114; *Angew. Chem. Int. Ed.* **2024**, *63*, e202316306).

As shown in **Figure R2**, the in-situ SEM tests demonstrated the WS₂ superstructure catalyst still maintained excellent mechanical properties after the stability test. The optical photographs in **Figure R3** show that the membrane electrodes of WS₂ superstructure do not display evident changes in appearance and color after the

durability test in AEM electrolyzer.

Additionally, in our original manuscript, we already provided the post-stability characterizations such as SEM, TEM, HAADF-STEM, corresponding EDS elemental mappings, and XRD of the WS₂ superstructure to investigate any structural changes after the long-term stability test in electrolyzer (**Supplementary Figs. 32–34**). As revealed in **Supplementary Figure 34**, the post-stability XRD spectra of WS₂ superstructure were similar to the XRD spectra of the pristine WS₂ superstructure, suggesting the well-maintained crystal structure after the stability test. Similarly, post-stability analysis of SEM, HRTEM, magnified HAADF-STEM and corresponding EDS elemental mappings confirmed that the caterpillar-like morphology of the catalyst is well-preserved and the W and S elements were homogeneous distributed within the whole WS₂ superstructure after the electrolyzer stability test (**Supplementary Figs. 32 and 33**).

Therefore, combining SEM, TEM, HAADF-STEM, EDS elemental mappings, XRD, XPS, ICP-OES, and in-situ SEM analysis, we conclude that the WS₂ superstructure as the cathode catalyst has excellent catalytic stability for HER catalysis in AEM electrolyzer.

Figure R1. XPS analysis of the WS₂ superstructure anode catalyst before and after AEM electrolyzer operation. (a) W 4f and (b) S 2p XPS signals of the WS₂ superstructure anode catalyst before and after HER operation in the AEM electrolyzer.

Table R1. The ICP-OES results of the electrolytes before and after stability tests of WS₂ superstructure catalyst.

Elements contents ($\mu\text{g mL}^{-1}$)	Before stability tests in AEM electrolyzer	After stability tests in AEM electrolyzer
W	< 0.1	0.21

Figure R2. In-situ SEM mechanical test of the WS₂ superstructure post-electrolysis in the AEM electrolyzer. Scale bar, 500 nm.

Figure R3. (a, b) Optical photographs of WS₂ superstructure membrane electrode before (a) and after (b) durability test for 1000 hours in the AEM electrolyzer.

We have added the **Figure R1** and **R2** as new **Supplementary Figure 35** and **36**, respectively, in revised Supplementary Information. The relevant discussions have been added into the revised manuscript (page 30, line 529–530).

3. The authors should perform a techno-economic analysis for the catalyst and provide a comparative picture with the current state-of-the-art electrocatalysts.

Response: We appreciate your professional suggestions. According to your suggestion, we investigated in detail the techno-economic analysis (TEA) derived from the U.S. Department of Energy's (DOE) Advanced Manufacturing Office (AMO) (<https://www.energy.gov/eere/iedo/life-cycle-assessment-and-techno-economic-analysis-training>). TEA is a method for evaluating the economic performance of a technology. A TEA assesses the overall value of a technology, allowing analysts to objectively weigh benefits against costs (*Nature* **2023**, 617, 724–729; *Nat. Sustain.* **2023**, 6, 827–837; *Nat. Sustain.* **2021**, 4, 911–919; *Nat. Catal.* **2019**, 2, 1062–1070). AMO's approach to TEA focuses on cost benchmarking. In a cost benchmarking approach, the costs of a new technology are compared head-to-head against those of the existing commercial technology that it would compete with in the marketplace. This is done to assess competitiveness. For successful commercialization, a new technology must be cost-competitive. The goal is cost parity (at least!) when comparing to a commercial benchmark, after accounting for technical performance benefits.

Manufacturing costs can be broken down into two main categories: Capital Expenses (CapEx) and Operating Expenses (OpEx) (**Figure R4**). CapEx are non-recurring costs such as equipment, buildings, and construction. In a TEA, these one-time facility costs are amortized over the assets' useful lifetime to relate CapEx to a specific production volume. OpEx are recurring costs such as materials, labor, and energy. These recurring costs can be either fixed or variable (Fixed operating expenses, like labor, do not change with production output; Variable operating expenses, like materials, are directly tied to production volume).

Figure R4. Schematic classification of manufacturing costs in TEA.

In the cost benchmarking approach, we evaluated the commercial viability of the new technology (WS₂ superstructure as cathode catalyst in electrolyzer for H₂ production) by comparing its manufacturing costs to those of a corresponding commercial benchmark (commercial Pt/C as cathode catalyst in electrolyzer for H₂ production). For all cost estimations, we assumed that the new technology has been deployed at a full industrial scale—even if its current technology readiness level is low. We defined the functional unit based on a quantitative measure of the intended function of the product. In other words, a catalyst can be defined by its ability to produce 1 kg of hydrogen at a low total cost.

A preliminary TEA was performed to estimate the CapEx, OpEx, and overall hydrogen production costs for an ideal 1 MW-scale anion-exchange membrane electrolyzer (AEMEL) plant, assuming complete performance retention from lab-scale tests to the plant. The boundaries of the TEA were set at the outlet of the AEMELs, i.e., hydrogen stocking and transportation costs have not been considered. The cost of the diaphragm/electrode package (DEP) for a single lab-scale cell (5 cm²) was calculated from the commercial price of each component or the price of its constituting raw materials (**Table R2**). Manufacturing costs related to the cathodes (i.e., deposition of the catalytic coating by ultrasonic spray technique) were not considered.

The CapEx of an ideal 1 MW-scale AEMEL plant based on the DEP configurations

tested at lab-scale was calculated starting from data provided by IRENA (<https://www.miningweekly.com/article/pgms-playing-atypical-role-in-alkaline-electrolyser-lowering-green-hydrogen-cost-outlook-2021-10-26>) and reports on currently operating large-scale water electrolytic plants (*Energy Environ. Sci.* **2021**, *14*, 6338–6348; *J. Environ. Chem. Eng.* **2021**, *9*, 106349) (**Table R2**). At first, the CapEx of a generic 1 MW-scale AEMEL plant was estimated from literature data (*J. Environ. Chem. Eng.* **2021**, *9*, 106349). Then, following the cost breakdown of a generic MW-scale AEMEL plant reported by IRENA and assuming AEMEL performances to meet those reported literature (*J. Environ. Chem. Eng.* **2021**, *9*, 106349; *EES Catal.* **2024**, *2*, 109–137; *Int. J. Hydrogen Energy* **2021**, *46*, 3379–3386; *Energy Convers. Manage.* **2024**, *302*, 118134), the CapEx was split into stack, DEP, and Balance of Plant (BOP) costs and relative subcomponents.

In brief, the overall cost breakdown of a typical MW scale AEMEL has been used to retrieve the system CapEx, including expenditures related to the BOP. Finally, by upscaling the size of the DEP (cathode, anode, and diaphragm) to 700 cm² and assuming the final ideal AEMEL to be composed of 5 stacks of 200-unit cells, the total CapEx of an ideal 1 MW-scale AEMEL plant for each DEP configuration tested in the present work was calculated. Such CapEx estimation was made under the assumption that cathode and anode costs scale linearly with plant size. The annual CapEx was retrieved from overall CapEx considering its depreciation through a capital recovery factor (CRF), calculated according to the following equation:

$$\text{CRF} = \frac{i_{\text{rate}} \times (1 + i_{\text{rate}})^n}{(1 + i_{\text{rate}})^n - 1}$$

where i_{rate} is the discount rate and n is the AEL plant lifetime (**Table R3**).

The OpEx of the plant was calculated starting from the data (i.e., current-voltage relationships) collected from our single cell AEMELs at lab scale. Several entries were considered to sum up to the overall system OpEx, namely: electricity fed to the AEMELs, process water, labor, maintenance, and other ancillary costs. The OpEx related to electricity and process water, which are dependent on the electrolyzer

performance, were calculated according to the gross power of the system and the water annual consumption, respectively (**Table R4**). Specifically, the cost of the electricity required for the actual electrolysis process was calculated according to the following equations:

$$I_{\text{total}} = I \times A_{\text{single cell-idea 1 MW-scale AEMEL}} \times n_{\text{cells per stack}} \times n_{\text{stacks per system}}$$

$$P_{\text{AEMEL (gross)}} = I_{\text{total}} \times E_{\text{cells}} \times t_{\text{annual AEMEL operation}}$$

$$\text{OpEx}_{\text{Electricity}} = P_{\text{AEMEL (gross)}} \times C_{\text{Electricity}}$$

where I (i) indicates the current (current density) flowing through the electrolyzer, A stands for area, E for voltage, P for power, t for time and C for cost. The expenses related to process water consumption have been estimated through the following equations:

$$m_{\text{H}_2\text{O consumed (per year)}} = m_{\text{produced H}_2 \text{ (per year)}} \times m_{\text{average H}_2\text{O consumption per kg of H}_2}$$

$$\text{OpEx}_{\text{H}_2\text{O}} = m_{\text{H}_2\text{O consumed (per year)}} \times C_{\text{H}_2\text{O}}$$

where m stands for mass.

On the other hand, labor, maintenance and other ancillary OpEx contributions were calculated as percentages of the total CapEx of the whole system. The total OpEx was computed summing up all the listed contributions and doubling the electricity-related expenses as BOP power consumption equals the AEMEL's one for plant scales superior to 1 MW.

Moreover, the amount of yearly produced H_2 ($\text{kg}_{\text{H}_2} \text{ year}^{-1}$) by the ideal AEMEL plant was calculated through the Faraday's law:

$$\text{annual H}_2 \text{ production} = \frac{I \times t \times FE \times \text{MM}_{\text{H}_2}}{n \times F}$$

where I is the total current delivered by the plant in 1 year, t is the time, FE is the

Faradaic efficiency, MM_{H_2} is the molecular mass of hydrogen ($g\ mol^{-1}$), n is the number of electrons transferred for each H_2 molecule generated (mol_e^-/mol_{H_2}) and F is the Faraday's constant ($C\ mol_e^{-1}$) (**Table R5**). Finally, the hydrogen production cost was calculated as:

$$H_2\ production\ cost\ (US\$/kg_{H_2}) = \frac{Annual\ CapEx + Annual\ OpEx}{Annual\ H_2\ production}$$

Based on the above analysis, we investigated the effect of the WS_2 superstructure as the cathode catalyst in AEMEL on the cost of produced hydrogen in comparison with the current state-of-the-art electrocatalysts, Pt/C ($70, 150$ and $300\ \mu g\ cm^{-2}$). **Figure R5** shows the impact of catalyst in the cathode on the CapEx, OpEx, and, overall hydrogen production costs as estimated by the TEA for an ideal 10-year-lifetime 1 MW AEMEL plant operating at single cell current density of $1\ A\ cm^{-2}$. In this case, the catalyst first affects the CapEx, which decreases by about $US\$0.03\ kg_{H_2}^{-1}$ when WS_2 superstructure is used as the cathode catalyst. In addition, the hydrogen production costs are mainly determined by OpEx, which, in turn, depends on the AEMEL performance. Thus, when the performance of non-precious metal catalysts is comparable to that of commercial precious metal catalysts, the use of non-precious metal catalysts reduces the overall cost of hydrogen production and results in profitability. The use of WS_2 superstructure catalyst resulted in a lower unit production cost of hydrogen ($US\$2.212\ kg_{H_2}^{-1}$) when compared to that of commercial Pt/C catalysts ($70\ \mu g\ cm^{-2}$: $US\$2.337\ kg_{H_2}^{-1}$; $150\ \mu g\ cm^{-2}$: $US\$2.274\ kg_{H_2}^{-1}$; $300\ \mu g\ cm^{-2}$: $US\$2.272\ kg_{H_2}^{-1}$), as shown in **Table R6**.

It is noted that the other factors beyond catalysts and AEMEL performance are determining hydrogen production cost, particularly the electricity price (*Nat. Nanotech.* **2019**, *14*, 1071–1074; *Science* **2020**, *368*, 1228–1233; *Angew. Chem. Int. Ed.* **2024**, e202319936; *Nat. Commun.* **2024**, *15*, 616). This analysis is outside of this report and will be the subject of further investigations.

In summary, this TEA study demonstrated an economically cost-effective option of hydrogen production based on as-prepared WS₂ superstructure as the cathode catalyst.

Figurer R5. CapEx, OpEx and overall hydrogen production cost for ideal 1 MW-scale AEMEL plants based on different cathode catalysts (Current density = 1 A cm⁻², AEMEL plant lifetime = 10 years).

Table R2. CapEx-related parameters assumed in the TEA.

Parameters of single cell CapEx calculation			
	Palatinum ^a	48.3	\$ g ⁻¹
	WS ₂ superstructure	0.26	\$ kg ⁻¹
DEP components	AvCarbMGL280 ^a	500	\$ m ⁻²
	D1021 Dispersion ^a (Nafion TM)	1.53	\$ g ⁻¹
	SSM ^b	11.2	\$ m ⁻²
	Sustainion® X37-50 Grade 60 ^b	511	\$ m ⁻²

Electrode area	Lab-scale AEMEL	5	cm ²
	Ideal 1 MW-scale AEMEL	700	cm ²
System configuration	n _{Cell} (cells per stack)	200	cells
	n _{Stacks} (stacks per system)	5	stacks
Parameters for breaking down the cost of a generic 1 MW-scale AEMEL plant			
System CapEx breakdown	Average stack cost (1 MW) ^c	270	\$ kW ⁻¹
	Stack CapEx share ^e	45	% of total CapEx
	BOP CapEx share ^e	55	% of total CapEx
System operative parameters ^d	Operative voltage	1.7	V
	Operative current density	1	A cm ⁻²
	Electrode area	700	cm ²
	Single cell power	1190	W

^a Ref. (<https://www.nrel.gov/docs/fy19osti/72740.pdf>). ^b Price from supplier. ^c Ref. (*J. Environ. Chem. Eng.* 2021, 9, 106349).

Table R3. Financial parameters assumed in the TEA.

Interest rate	4.5	%
Plant lifetime ^a	10	years
Capital recovery factor	0.126	-
Electricity cost ^b	0.02	\$ kWh ⁻¹

^a Average plant lifetime for MW-scale electrolyzer (U.S. Department of Energy Hydrogen Program Plan, <https://www.hydrogen.energy.gov/pdfs/hydrogen-program-plan-2020.pdf>).

^b Target price of renewable electric energy (Green Hydrogen Cost Reduction-Scaling up electrolyzers to meet the 1.5°C climate goal. Int. Renew. Energy Agency (IRENA), 2020. <https://irena.org/publications/2020/Dec/Green-hydrogen-cost-reduction>).

Table R4. OpEx-related parameters assumed in the TEA.

OpEX _{Process water}	Process water usage ^a	10	L kg _{H₂} ⁻¹
	Process water cost ^b	0.0014	\$ L _{H₂} ⁻¹
Other OpEx entries ^a	OpEX _{Labour}	0.3	% of total CapEx
	OpEX _{Maintenance}	2.5	% of total CapEx
	OpEX _{Ancillary}	1	% of total CapEx
-	Time of operation	8400	h year ⁻¹

^a Ref. (*J. Environ. Chem. Eng.* 2021, 9, 106349). ^b Ref. (*Cell Reports Phys. Sci.* 2020, 1, 100209).

Table R5. Electrochemical and process-related parameters assumed in the TEA.

Faradaic efficiency	100	%
Number of exchanged electrons	2	mol _{e⁻} mol _{H₂} ⁻¹
Faraday's constant	96485	C mol _{e⁻} ⁻¹
H ₂ molar mass	0.02	g mol ⁻¹
HHV _{H₂} (hydrogen higher heating value)	141.7	kJ g _{H₂} ⁻¹
LHV _{H₂} (hydrogen lower heating value)	120.0	kJ g _{H₂} ⁻¹

Table R6. Techno-economic model parameters based on different cathode catalysts.

Catalyst	CapEx (\$ year ⁻¹)	OpEx (\$ year ⁻¹)	CapEx/OpEx ratio	H ₂ production cost (\$ kg _{H₂} ⁻¹)
WS ₂ superstructure	74238	414910	0.178	2.212
Pt/C (70 μg cm ⁻²)	78973	437805	0.180	2.337
Pt/C (150 μg cm ⁻²)	79329	423454	0.187	2.274
Pt/C (300 μg cm ⁻²)	79996	422544	0.189	2.272

We truly thank you for the insightful comments and kind suggestions again! The reply for each question/comment is expected to reach the high criteria for you.

Reply to Reviewer 2 and revisions made accordingly:

Electrocatalytic hydrogen evolution reaction (HER) is an important reaction for energy storage/conversion. Actually, it is a bottleneck to produce hydrogen energy economy through water splitting. Therefore, an efficient HER electrocatalyst is critical for development of hydrogen electrochemistry. In this work, author reported WS₂ superstructure with highly catalytic property of hydrogen evolution in alkaline solution. The as-synthesized WS₂ superstructure can produce industrial-level current density for hydrogen generation. Remarkably, the AEM electrolyser using WS₂ superstructure cathode achieves an ultralow cell voltage of 1.70 V at 1 A cm⁻² under industrial testing (30 wt% KOH at 60°C). However, similar works on WS₂ HER electrocatalysts have been published, such as Nat. Mater., 2013, 12, 850–855; ACS Nano 2019, 13, 10448–10455; Energy Environ. Sci., 2014, 7, 2608-2613; Nat. Commun., 2021, 12, 709; Small, 2020, 16, 1905000, and so on. Therefore, this work lacks enough novelty, and the manuscript does not bring a compelling scientific case that connects structural characterizations and electrochemical findings. Therefore, the manuscript doesn't meet the high standard of Nature Communication in its present state. The current submission can be further improved and published elsewhere if the authors can address the following issue.

Response: We sincerely thank you for carefully reviewing our manuscript and providing valuable comments. We have studied all comments carefully and have made corrections which we sincerely hope meet with approval. As will be shown below, we have carried out a series of supplemented experiments and detailed analysis to further connect structural characterizations and electrochemical findings of this work following your constructive suggestions.

Before answering the specific questions remarked, we would like to firstly address the question regarding the novelty of the manuscript. We thank your comments on the novelty of our work, and we are pleased to clarify this issue. We agree that similar works on WS₂ HER electrocatalysts were being published in the last few years.

However, evaluating the impact/novelty of work should reply on what it has done to solve the remaining scientific or engineering challenges/questions of the studied materials or reactions, not only on the number of published papers. To further highlight and demonstrate the innovativeness of our work, we summarize the main achievements of the present study as follows:

(1) Earth-abundant tungsten disulfide (WS_2)-based material, which is regarded as a promising HER catalyst to replace noble Pt catalysts, is such an example proved by both experimental and theoretical studies (*J. Am. Chem. Soc.* **2005**, *127*, 5308–5309; *Nat. Mater.* **2013**, *12*, 850–855). As the reviewer mentioned, previous studies have demonstrated excellent HER performance of several WS_2 electrocatalysts (*Nat. Mater.* **2013**, *12*, 850–855; *ACS Nano* **2019**, *13*, 10448–10455; *Energy Environ. Sci.* **2014**, *7*, 2608–2613; *Nat. Commun.* **2021**, *12*, 709; *Small*, **2020**, *16*, 1905000). However, these reported WS_2 -based catalysts are utilized only in acid media. In effect, the kinetics of WS_2 electrocatalyst becomes rather sluggish in more practically viable base solution, which, in essence, is mainly attributed to the intrinsic structural feature of WS_2 , resulting in very high kinetic energy barrier of the initial water dissociation process and the strong adsorption of the formed -OH on its surface. (*Angew. Chem. Int. Ed.* **2024**, *63*, e202316306; *Angew. Chem. Int. Ed.* **2018**, *57*, 7568–7579). This is a critical issue that an efficient alkaline electrocatalyst should reach a balance between catalyst-hydrogen and catalyst-hydroxyl interactions, which is the main difference from the acidic HER process.

Aiming at this issue, our approach to obtaining edge-stepped defects can modulate the coordination environment and electronic structure of the WS_2 superstructure at the alkaline HER, boosting the intrinsic HER kinetics. The excellent alkaline HER activity of the WS_2 superstructure catalyst can be achieved by a spatially decoupled water dissociation and hydrogen desorption mechanism, where the edge-stepped defects accelerate the water dissociation rate (low energy barrier of water dissociation), and the generated H intermediates would then migrate to the sulfur atoms and recombine to have H_2 evolution (optimal hydrogen adsorption energy). This enables the seamless

integration of water dissociation with the adsorption and desorption processes of hydrogen on the WS₂ superstructure, yielding extraordinary alkaline HER properties relative to commercial Pt/C (20 wt%) (**Figure R6** and **Table R7**) and other WS₂ electrocatalysts reported. In addition, in-situ surface-enhanced Raman spectroscopy (SERS) characterizations uncover the origin of the superior activity and establish a structure-performance relationship, that is, under HER conditions, the real active sites are the unsaturated sulfur atoms on the edge-stepped defects. Meanwhile, the WS₂ superstructure exhibits the edge-rich characteristic, as evidenced by **Figure R8**, which can provide abundant unsaturated sulfur atoms as catalytic active sites for HER.

This success represents a breakthrough in WS₂ electrocatalysts for improving the sluggish kinetics for electrochemical reduction of water to molecular hydrogen in alkaline environments. Thus, our approach to obtain edge-stepped defects for modulating the intrinsic activity of WS₂ for achieving high-performance alkaline HER catalysis is conceptually novel and can also be extended to the other fields of materials science and electrocatalysis.

(2) More importantly, the three-dimensional (3D) superstructure designed for operation at high current density as required by industrial-relevant electrolyzer is the significant innovation of this work. As we have mentioned in our introduction, anion-exchange membrane water electrolysis (AEMWE) combines the advantages of alkaline water electrolysis (ALKWE) and proton exchange membrane water electrolysis (PEMWE), and is attracting increasing attention for its potential to realize low-cost and high-performance hydrogen production. For efficient H₂ production in AEM electrolyzer, it is vital for the community to undertake the essential task of developing active and stable non-precious metal electrocatalysts for the alkaline HER. These catalysts must meet the specific demands of industrial AEMWE while exhibiting low overpotential and prolonged lifetime at an ampere-level current density. The current limitations of WS₂-based catalysts in AEMWE lie in their inferior ability of water dissociation and the deactivation (poisoning) of catalytic active sites during prolonged operation at industrial-level current density, as well as the significantly reduced lifetime

attributed to mechanical abrasion resulting from high-density gas-liquid exchanges. For the reference papers mentioned by the Reviewer 2, all these reported WS₂-based catalysts are utilized only in a conventional three-electrode system at low current densities (e.g., 10 mA cm⁻²) in the laboratory.

Aiming at the above issue, we demonstrate an efficient alkaline HER strategy at industrial-level current density wherein a flexibly deformable WS₂ superstructure was designed to serve as the cathode catalyst for AEM water electrolysis. The superstructure features bond-free van der Waals interaction among the low Young's modulus nanosheets to ensure excellent mechanical flexibility, as well as a stepped edge defect structure of nanosheets to endow high catalytic activity and a favourable reaction interface micro-environment. The unique flexible WS₂ superstructure can effectively withstand the impact of high-density gas-liquid exchanges and facilitate mass transfer, endowing exceptional long-term durability under industrial-scale current density. For the first time, we applied this well-designed WS₂ superstructure catalysts to AEM water electrolyzers and achieved excellent performance. The AEM electrolyzer using WS₂ superstructure cathode achieves ultralow cell voltage of 1.70 V at 1 A cm⁻² under industrial testing conditions (30% KOH at 60°C), while maintaining stable electrolysis for 1000 h at 1 A cm⁻² with apparent degradation rate only at 9.67 μV h⁻¹. This decay rate means that the cell voltage would suffer a voltage increase as low as 0.235 V over a 1-year-long operation. In addition, the techno-economic analysis (TEA) study demonstrated an economically cost-effective option of hydrogen production based on as-prepared WS₂ superstructure as the cathode catalyst (**Figure R5, Table R2–6**). Our design concept and mechanistic analysis of WS₂ superstructure as an active and stable cathode catalyst at high current density is an advancement in the field of non-noble-metal alkaline water electrolysis, which is a step forward towards the development of large-scale AEM electrolyzers.

We wish that the reviewer could share our confidence and belief that the work reported in this paper deserves to be published in *Nature Communications*.

1. This paper seems to list only the electrocatalytic properties of WS₂ catalysts in its present state. Based geometric area normalized LSVs, WS₂ catalysts exhibit a superior HER performance relative to Pt/C. However, ECSA-normalized current density and turnover frequency (TOF) of WS₂ and Pt/C is also compared.

Response: Thanks for the comments. According to your suggestion, we have added the ECSA-normalized current density and turnover frequency (TOF) of WS₂ superstructure and Pt/C for better comparison of electrocatalytic activity. To calculate the ECAS and per-site TOF for the WS₂ superstructure and Pt/C catalyst, we adopt the method applied by Jaramillo et al., where cycle voltammetry (CV) method was used in this part (*J. Am. Chem. Soc.* **2013**, *135*, 16977–16987; *Angew. Chem. Int. Ed.* **2014**, *53*, 14433–14437; *J. Am. Chem. Soc.* **2015**, *137*, 4347–4357; *Energy Environ. Sci.* **2015**, *8*, 3022–3029; *Nat. Commun.* **2019**, *10*, 2650) (Detailed calculation method was presented in **Supplementary Note 2** in Supplementary Information). As shown in **Figure R6**, the ECSA-normalized current density demonstrated that the WS₂ superstructure still reveals substantially higher HER current density than that of Pt/C at a high overpotential (> -228 mV vs. RHE). The TOF results in **Table R7** demonstrated that the TOF (at -200 mV vs. RHE) of WS₂ superstructure is 4.011 s⁻¹, higher than that of commercial Pt/C (2.670 s⁻¹). These results demonstrated that the WS₂ superstructure catalysts exhibit a superior HER performance relative to Pt/C (20 wt%).

Figure R6. (a) HER activity normalized for the electrochemical active surface area (ECSA). (b) Comparison of the ECSA and J_{ECSA} (at -300 mV vs. RHE) of WS₂

superstructure and commercial Pt/C (20 wt%).

Table R7. The ECSA, TOF, and ECSA-normalized current density values of different catalysts.

Catalyst	ECSA (cm ²)	TOF (s ⁻¹) @overpotential (mV)	J_{ECSA} ($\eta = -300$ mV vs. RHE) (mA cm ⁻²)
WS ₂ superstructure	320.8	0.600@100 4.011@200	3.53
Pt/C (20 wt%)	191.4	0.940@100 2.670@200	2.61

We have added the **Figure R6** and **Table R7** as new **Supplementary Figure 23** and **Supplementary Table 2**, respectively, in revised Supplementary Information.

2. The determination of active centers should be further discussed based on more experiments and DFT calculations. In-situ XRD and Raman measurements should be carried out to confirm the real active sites for HER.

Response: Thanks for your comments and suggestions. It is difficult for us to determine the active centers for HER by in-situ XRD, because it is beyond the limit of our XRD instrument. According to your suggestion, we have carried out a series of in-situ Raman combined with electrochemical measurements to confirm the real active sites for HER. In addition, we have added more discussion about the active centers based on the experiments and DFT calculations (page 22, line 387–406).

Capturing direct spectroscopic evidence of intermediates produced during catalytic processes is key to unraveling the real active sites for HER (*Nat. Energy* **2019**, *4*, 60–67; *Nat. Commun.* **2016**, *7*, 12440). Surface-enhanced Raman spectroscopy (SERS)

can provide surface-sensitive as well as chemical bond specific signals at the atomic level, making it a powerful fingerprint spectroscopy which can in-situ identify the active sites as well as the surface reaction intermediates during catalytic processes (*Nature* **2021**, *600*, 81–85; *Nat. Commun.* **2023**, *14*, 5289; *Nat. Mater.* **2019**, *18*, 697–701). The in-situ SERS was used to monitor the HER process in 1.0 M KOH electrolyte.

As shown in **Figure R7**, during the potential going down from 0 to –200 mV, the broad peak spanning from 3000 to 3700 cm^{-1} corresponds to the adsorbed water peaks in the WS₂ superstructure sample (*Adv. Mater.* **2022**, *34*, 2110604; *Angew. Chem. Int. Ed.* **2020**, *59*, 22397). When the potential reaches –50 mV, an obvious Raman peak located at ca. 2550 cm^{-1} appears and its intensity increases as the potential further decreases to –200 mV. The band detected at 2550 cm^{-1} is ascribed to the stretching vibration of S-H bond, $\nu(\text{S-H})$ (*ACS Catal.* **2016**, *6*, 7790–7798; *J. Am. Chem. Soc.* **2020**, *142*, 7161–7167), indicating that the H atom is directly bonded to the sulfur atom of WS₂ superstructure during the HER. Importantly, it is worth mentioning that the vibrational signals, $\nu(\text{W-H})$, between 1831 and 1993 cm^{-1} (*J. Chem. Soc., Dalton Trans.* **1972**, 2492–2496; *J. Phys. Chem. A* **2002**, *106*, 6720–6729) have not been observed in the in-situ SERS measurements (**Figure R7**).

Based on detailed fingerprint information mentioned above, it is concluded that the sulfur atom of WS₂ is the catalytic active site for the HER. Albeit different from our WS₂ superstructure, the sulfur atoms in amorphous MoS_x (*ACS Catal.* **2016**, *6*, 7790–7798) and MoS₂ nanosheets (*J. Am. Chem. Soc.* **2020**, *142*, 7161–7167) were also confirmed to be the active sites for HER.

Figure R7. In-situ SERS spectra of the WS₂ superstructure catalyst at various potentials (vs. RHE) under HER conditions in 1.0 M KOH.

According to the previous report (*Faraday Discuss.* **2008**, *140*, 219–231; *Sci. Adv.* **2015**, *1*, e1500259; *ACS Catal.* **2016**, *6*, 861–867), the irreversible electrochemical oxidation of metal sulfides was also investigated as a measure of the real active sites of WS₂ superstructure catalysts. Irreversible electrochemical oxidations were performed on both samples (WS₂ superstructure and conventional WS₂ nanosheets, as shown in **Figure R8a**) using 0.5 M H₂SO₄ as the electrolyte at a scan rate of 60 mV s⁻¹. **Figure R8b** shows the cyclic voltammetry (CV) curves of the WS₂ superstructure and conventional WS₂ nanosheets. A peak centered at ~1.05 V (vs. RHE) in the CV curve of conventional WS₂ nanosheets can be ascribed to the oxidation of sulfur atoms on the basal planes of WS₂ (*Faraday Discuss.* **2008**, *140*, 219–231). In contrast, one apparent oxidation peak centered at ~0.67 V vs. RHE was observed in the CV of the WS₂ superstructure, possibly due to its edge-rich (including both sheet edges and stepped defect edges) features, as sulfur atoms on the edges of WS₂ nanostructures are expected to be more readily oxidized than does the basal plane (*Sci. Adv.* **2015**, *1*, e1500259; *ACS Catal.* **2016**, *6*, 861–867).

Figure R8. (a) TEM images of WS₂ superstructure (left) and conventional WS₂ nanosheets (right). The conventional WS₂ nanosheets were synthesized according to a previous report for comparison (*Nat. Commun.* **2021**, *12*, 5070; *J. Am. Chem. Soc.* **2014**, *136*, 14121–14127; *Nano Energy* **2018**, *50*, 176–181). (b) Irreversible electrochemical oxidation cyclic voltammograms. Irreversible electrochemical oxidation of WS₂ superstructure and conventional WS₂ nanosheets in 0.5 M H₂SO₄ at a scan rate of 60 mV s⁻¹.

To further verify the significance of S active sites for HER, zinc nitrate (Zn(NO₃)₂) was employed in the poisoning experiment to block the S sites, as depicted in **Figure R9**, according to the previous report (*Chem* **2017**, *3*, 122–133; *Angew. Chem. Int. Ed.* **2024**, e202401453; *Nat. Commun.* **2019**, *10*, 5231). As shown in **Figure R10a**, the HER activity of the WS₂ superstructure decreased significantly after being soaked in 1.0 mM

Zn(NO₃)₂ aqueous solution for 30 s. The poisoning of active S sites resulted in an increase of overpotential from 52 mV to 201 mV at 10 mA cm⁻² and 89 to 386 mV at 50 mA cm⁻² (**Figure R10b**). We can conclude from the poison experiments that edge-stepped defects with abundant unsaturated sulfur atoms play an important role in determining the HER property of the WS₂ superstructure electrocatalyst. During the HER process, these stepped-like structures could provide abundant unsaturated sulfur atoms along with their curved edge profiles to serve as active centers to promote the evolution of H species.

Based on these results, it is confirmed that the edge-rich characteristic of the WS₂ superstructure and the unsaturated sulfur atoms on the edges are the real active sites of the WS₂ superstructure.

Figure R9. Schematic diagram of the active site blocking mechanism.

Figure R10. HER polarization curves (a) and the required overpotential at 10 and 50

mA cm^{-2} (b) of WS_2 superstructure before and after soaking in 1.0 mM $\text{Zn}(\text{NO}_3)_2$ solution for 30 s.

We have added the **Figure R7–10** as **Supplementary Figure 22–24** in the revised Supplementary Information. Some necessary discussion has been inserted into our revised manuscript, see page 22 lines 387–406. For your convenience, the added discussion are as follows:

“To understand the underlying factors contributing to the excellent activity of the WS_2 superstructure, in-situ surface-enhanced Raman spectroscopy (SERS) was performed to analyze surface reaction intermediates during the HER process⁶⁸. As shown in Supplementary Fig. 22, the broad peak spanning from 3000 to 3700 cm^{-1} corresponds to the adsorbed water peaks in the WS_2 superstructure sample, demonstrating its excellent water adsorption capacity⁶⁹. This result is in good consistent with the DFT calculations (Fig. 2j). The successful observation of S-H bonds ($\sim 2550 \text{ cm}^{-1}$) via in-situ SERS directly confirmed that sulfur atoms serve as the catalytic active sites in WS_2 superstructure for HER^{70–72}. In addition, the irreversible electrochemical oxidation study demonstrated the edge-rich characteristic of the WS_2 superstructure (Supplementary Fig. 23)⁷³. The poison experiments using zinc nitrate further demonstrated that the edge-stepped defects with abundant unsaturated sulfur atoms play an important role in determining the HER property of the WS_2 superstructure electrocatalyst (Supplementary Fig. 24)^{70,74}. We can conclude from the in-situ SERS, irreversible electrochemical oxidation, and poison experiments that edge-stepped defects with abundant unsaturated sulfur atoms play an important role in determining the HER property of the WS_2 superstructure electrocatalyst. During the HER process, these stepped-like structures could provide abundant unsaturated sulfur atoms along with their curved edge profiles to serve as active centers to promote the evolution of H species”

[68] Wang, Y.-H., Zheng, S., Yang, W.-M., Zhou, R.-Y., He, Q.-F., Radjenovic, P., Dong, J.-C., Li, S., Zheng, J., Yang, Z.-L., Attard, G., Pan, F., Tian, Z.-Q., Li, J.-F. In situ

- Raman spectroscopy reveals the structure and dissociation of interfacial water. *Nature* **600**, 81-85 (2021).
- [69] Wang, J., Yang, T., Li, X., Zhang, H., Zhang, Y., He, Y., Xue, H. Hydrogen evolution reaction activity enhancement from active site turnover mechanism. *J. Energy Chem.* **92**, 629-638 (2024).
- [70] Yang, C., Yue, J., Wang, G., Luo, W. Activating and identifying the active site of RuS₂ for alkaline hydrogen oxidation electrocatalysis. *Angew. Chem. Int. Ed.*, e202401453 (2024).
- [71] Chen, J., Liu, G., Zhu, Y.-z., Su, M., Yin, P., Wu, X.-j., Lu, Q., Tan, C., Zhao, M., Liu, Z., Yang, W., Li, H., Nam, G.-H., Zhang, L., Chen, Z., Huang, X., Radjenovic, P.M., Huang, W., Tian, Z.-q., Li, J.-f., Zhang, H. Ag@MoS₂ core-shell heterostructure as SERS platform to reveal the hydrogen evolution active sites of single-layer MoS₂. *J. Am. Chem. Soc.* **142**, 7161-7167 (2020).
- [72] Guo, S., Li, Y., Tang, S., Zhang, Y., Li, X., Sobrido, A.J., Titirici, M.M., Wei, B. Monitoring hydrogen evolution reaction intermediates of transition metal dichalcogenides via operando Raman spectroscopy. *Adv. Funct. Mater.* **30**, 2003035 (2020).
- [73] Miao, J., Xiao, F.-X., Yang, H.B., Khoo, S.Y., Chen, J., Fan, Z., Hsu, Y.-Y., Chen, H.M., Zhang, H., Liu, B. Hierarchical Ni-Mo-S nanosheets on carbon fiber cloth: A flexible electrode for efficient hydrogen generation in neutral electrolyte. *Sci. adv.* **1**, e1500259 (2015).
- [74] Hu, C., Ma, Q., Hung, S.-F., Chen, Z.-N., Ou, D., Ren, B., Chen, H.M., Fu, G., Zheng, N. In situ electrochemical production of ultrathin nickel nanosheets for hydrogen evolution electrocatalysis. *Chem* **3**, 122-133 (2017).

3. Beside from the kinetics, author should explain why WS₂ catalysts show an extraordinary HER property compared to other WS₂ electrocatalysts reported.

Response: We appreciate this valuable comment and are pleased to clarify this issue. Earth-abundant tungsten disulfide (WS₂)-based material, which is regarded as a

promising HER catalyst to replace noble Pt catalysts, is such an example proved by both experimental and theoretical studies (*J. Am. Chem. Soc.* **2005**, *127*, 5308–5309; *Nat. Mater.* **2013**, *12*, 850–855). Nevertheless, despite extensive efforts in recent years (*Nat. Mater.* **2013**, *12*, 850–855; *Energy Environ. Sci.* **2014**, *7*, 2608–2613; *Adv. Mater.* **2015**, *27*, 4234–4241; *Adv. Mater.* **2018**, *30*, 1705509; *ACS Nano* **2019**, *13*, 10448–10455; *Small* **2020**, *16*, 1905000; *Nano Energy* **2018**, *50*, 176–181; *Angew. Chem. Int. Ed.* **2021**, *60*, 21550–21557; *Nat. Commun.* **2021**, *12*, 709; *J. Am. Chem. Soc.* **2022**, *144*, 4863–4873), its high HER activity is only limited to acidic media, and the kinetics become rather sluggish in more practically viable base solution, which, in essence, is mainly attributed to the intrinsic structural feature of WS₂, resulting in very high kinetic energy barrier of the initial water dissociation process and the strong adsorption of the formed -OH on its surface (*Angew. Chem. Int. Ed.* **2024**, *63*, e202316306; *Angew. Chem. Int. Ed.* **2018**, *57*, 7568–7579). This is a critical issue that an efficient alkaline electrocatalyst should reach a balance between catalyst-hydrogen and catalyst-hydroxyl interactions, which is the main difference from the acidic HER process.

To explain why WS₂ catalysts show an extraordinary HER property compared to other WS₂ electrocatalysts reported, we summarize the three main points as follows:

Firstly, the excellent alkaline HER activity of the 3D WS₂ superstructure catalyst can be understood by a spatially decoupled water dissociation and hydrogen desorption mechanism, where the edge-stepped defects accelerate the water dissociation rate (low energy barrier of water dissociation), and the generated H intermediates would then migrate to the sulfur atoms and recombine to have H₂ evolution (optimal hydrogen adsorption energy). This enables the seamless integration of water dissociation with the adsorption and desorption processes of hydrogen on the WS₂ superstructure, yielding extraordinary alkaline HER properties relative to commercial Pt/C (20 wt%) (**Figure R6**) and other WS₂ electrocatalysts reported.

Secondly, in-situ surface-enhanced Raman spectroscopy (SERS) characterizations uncover the origin of the superior activity and establish a structure-performance relationship, that is, under HER conditions, the real active sites are the unsaturated

sulfur atoms on the edge-stepped defects (**Figure R7**). Meanwhile, the 3D WS₂ superstructure exhibits the edge-rich characteristic, as evidenced by **Figure R8**, which can provide abundant unsaturated sulfur atoms as catalytic active sites for HER.

Finally, the 3D WS₂ superstructure provided an unobstructed mass diffusion pathway to ensure enough reactants reaching the catalytic active sites and in-time release of as-obtained intermediates and productions. Besides, the unique flexible WS₂ superstructure can effectively withstand the impact of high-density gas-liquid exchanges and facilitate mass transfer, endowing exceptional long-term durability under industrial-scale current density.

These three reasons are the main contributors to the extraordinary alkaline HER properties of the WS₂ superstructure over other WS₂ electrocatalysts reported.

Some necessary discussion has been inserted into our revised manuscript to explain why WS₂ superstructure shows an extraordinary HER property compared to other WS₂ electrocatalysts reported, see page 22 lines 387–406.

4. The authors stated edge-stepped defects plays an important role in determining the HER property of WS₂ electrocatalyst. However, there is no experiment results to confirm it. How to control the edge-stepped defects in WS₂ electrocatalyst.

Response: We sincerely appreciate your valuable comment. Both theoretical (*J. Am. Chem. Soc.* **2005**, *127*, 5308–5309) and experimental (*Science* **2007**, *317*, 100–102; *Nat. Chem.* **2018**, *10*, 1246–1251) studies concluded that the HER activity arises from the sites located along the edges of the two-dimensional (2D) transition metal dichalcogenides (TMDs) layers, while the basal surfaces are catalytically inert. That is, the unsaturated sulfur atoms on the edges play a crucial role in HER catalysis. In our work, the introduction of edge-stepped defects in WS₂ superstructure could expose a large number of W-S dangling bonds and unsaturated sulfur atoms as the active sites for HER.

Based on the in-situ surface-enhanced Raman spectroscopy (SERS) measurement (**Figure R11**), we confirmed that the sulfur atom of WS₂ superstructure is the real

catalytic active site for the HER.

Figure R11. In-situ SERS spectra of the WS₂ superstructure catalyst at various potentials (vs. RHE) under HER conditions in 1.0 M KOH.

Simultaneously, besides the HRTEM images (**Figure 2f** and **Supplementary Figure 9**), the irreversible electrochemical oxidation of metal sulfides was also investigated as a measure of their edge sites. Irreversible electrochemical oxidations were performed on both samples (WS₂ superstructure and conventional WS₂ nanosheets, as shown in **Figure R12a**) using 0.5 M H₂SO₄ as the electrolyte at a scan rate of 60 mV s⁻¹. **Figure R12b** shows the cyclic voltammetry (CV) curves of the WS₂ superstructure and conventional WS₂ nanosheets. A peak centered at ~1.05 V (vs. RHE) in the CV curve of conventional WS₂ nanosheets can be ascribed to the oxidation of basal planes of WS₂ (*Faraday Discuss.* **2008**, *140*, 219–231). In contrast, one apparent oxidation peak centered at ~0.67 V vs. RHE was observed in the CV of the WS₂ superstructure, possibly due to its edge-rich (including both sheet edges and stepped defect edges) features, as edges of WS₂ nanostructures are expected to be more readily oxidized than does the basal plane (*Sci. Adv.* **2015**, *1*, e1500259; *ACS Catal.* **2016**, *6*, 861–867). Therefore, the edge-rich characteristic of the WS₂ superstructure is confirmed again.

Figure R12. (a) TEM images of WS₂ superstructure (left) and conventional WS₂ nanosheets (right). The conventional WS₂ nanosheets were synthesized according to a previous report for comparison (*Nat. Commun.* **2021**, *12*, 5070; *J. Am. Chem. Soc.* **2014**, *136*, 14121–14127; *Nano Energy* **2018**, *50*, 176–181). (b) Irreversible electrochemical oxidation cyclic voltammety curves. Irreversible electrochemical oxidation of WS₂ superstructure and conventional WS₂ nanosheets in 0.5 M H₂SO₄ at a scan rate of 60 mV s⁻¹.

To verify the significance of S active sites for HER, zinc nitrate (Zn(NO₃)₂) was employed in the poisoning experiment to block the S sites, as depicted in **Figure R13**, according to the previous report (*Chem* **2017**, *3*, 122–133; *Angew. Chem. Int. Ed.* **2024**, e202401453; *Nat. Commun.* **2019**, *10*, 5231). As shown in **Figure R14a**, the HER activity of the WS₂ superstructure decreased significantly after being soaked in 1.0 mM

Zn(NO₃)₂ aqueous solution for 30 s. The poisoning of active S sites resulted in an increase of overpotential from 52 mV to 201 mV at 10 mA cm⁻² and 89 to 386 mV at 50 mA cm⁻² (**Figure R14b**). We can conclude from the poison experiments that edge-stepped defects with abundant unsaturated sulfur atoms play an important role in determining the HER property of the WS₂ superstructure electrocatalyst. During the HER process, these stepped-like structures could provide abundant unsaturated sulfur atoms along with their curved edge profiles to serve as active centers to promote the evolution of H species.

Figure R13. Schematic diagram of the active site blocking mechanism.

Figure R14. HER polarization curves (a) and the required overpotential at 10 and 50 mA cm⁻² (b) of WS₂ superstructure before and after soaking in 1.0 mM Zn(NO₃)₂ solution for 30 s.

In our work, the control of edge-stepped defects in WS₂ electrocatalyst mainly involves the reaction time of the precursor. Through extending the synthesis time, WS₂ superstructures with different concentrations of edge-stepped defects were successfully synthesized, denoted as WS₂-48h, WS₂-60h and WS₂-72h. It is noteworthy that all of the above WS₂ samples exhibit the superstructural characteristics we have described. The electron paramagnetic resonance (EPR) was performed to evaluate the concentrations of edge-stepped defects in WS₂ superstructure with different reaction times. The signature of the W-S dangling bonds can be detected at $g = 2.003$ (**Figure R15**). The intensity of the signal is weakened during the increased synthesis time, implying the gradually decreased concentration of the edge-stepped defects in WS₂ superstructure sample.

Figure R15. EPR spectra of WS₂-48h, WS₂-60h, and WS₂-72h samples.

We have added the **Figure R11–14** as **Supplementary Figure 22–24** in the revised Supplementary Information. Some necessary discussion has been inserted into our revised manuscript, see page 22 lines 387–406.

5. On the one hand, the authors think edge-stepped defects can improve the WS₂ electrocatalyst. On the other hand, the authors explored the effect of mass transfer behavior of WS₂ electrocatalyst on promoting the HER property.

Response: Thank you very much for this comment. We appreciate the valuable comments and are pleased to clarify this issue. Generally speaking, gas-involving electrocatalysis occurs only at the interface of solid electrocatalyst, liquid electrolyte, and gaseous reactants/products, where electrons and ions/molecules can contact at catalytic active sites, so-called triple-phase boundary regions. In this framework, for the electrocatalytic hydrogen evolution reaction (HER), three critical steps are sequentially coupled with each other and strongly affect HER performance (*Sci. Adv.* **2023**, *9*, eadd6978; *Accounts Chem. Res.* **2018**, *51*, 881–889), as schematized in **Figure R16**: (1) mass diffusion in the electrolyte toward and from the electrochemical surface; (2) electron transfer from the conducting support to active sites and then redox intermediates; and (3) surface reaction involving the adsorption of reactants, interfacial charge transfer, and the desorption of intermediates/products at active centers.

Figure R16. Schematic representation of the gas-involving electrocatalysis. Three critical steps are coupled with each other, including (1) mass diffusion, (2) electron transfer, and (3) surface reaction.

Throughout the process, the surface reaction process is mainly controlled by the intrinsic electrocatalytic activity of the active sites which was typically demonstrated by the corresponding volcano-shaped plot (*Chem. Soc. Rev.* **2015**, *44*, 2060–2086; *Nat.*

Energy **2016**, *1*, 16130; *Nat. Chem.* **2012**, *4*, 873–886; *ACS Catal.* **2014**, *4*, 3957–3971), while other factors including mass diffusion, active site exposure, electron transfer which are dominated by the gas/liquid/solid contact, can greatly influence the overall activity of the catalysts (*J. Am. Chem. Soc.* **2017**, *139*, 12402–12405; *Accounts Chem. Res.* **2018**, *51*, 1590–1598; *J. Am. Chem. Soc.* **2020**, *142*, 1857–1863; *Adv. Mater.* **2024**, *36*, 2307925; *Nat. Commun.* **2024**, *15*, 2346). For example, even poor performance would be obtained on the terrific electrocatalysts whose existed active sites with high intrinsic activity are not fully utilized. In particular, in water splitting industry, a great number of gas bubbles will generate on the electrode surface as large current densities are usually required. In this case, if the electrode shows inferior mass transfer behavior, a large number of gas bubbles will gather around the surface and block the diffusion of electrolyte, resulting in huge reaction resistance and the so-called “dead zone”, as shown in **Figure R17**. Consequently, the electrocatalytic HER performance depends not only on the intrinsic activity of the electrocatalysts but also on the rational design and construction of the solid-liquid-gas three-phase interfaces to maximize intrinsic activity of the active sites and facilitate efficient electrode reaction process.

Figure R17. Large number of big H₂ bubbles are generated from the electrodes and gather around surface during HER.

In this article, firstly, the excellent alkaline HER activity of the WS₂ superstructure catalyst can be understood by a spatially decoupled water dissociation and hydrogen desorption mechanism, where the edge-stepped defects accelerate the water dissociation rate (low energy barrier of water dissociation), and the generated H

intermediates would then migrate to the sulfur atoms and recombine to have H₂ evolution (optimal hydrogen adsorption energy). This enables the seamless integration of water dissociation with the adsorption and desorption processes of hydrogen on the WS₂ superstructure, yielding extraordinary alkaline HER properties relative to commercial Pt/C (20 wt%) (**Figure R6** and **Table R7**) and other WS₂ electrocatalysts reported.

Secondly, in-situ surface-enhanced Raman spectroscopy (SERS) characterizations uncover the origin of the superior activity and establish a structure-performance relationship, that is, under HER conditions, the real active sites are the unsaturated sulfur atoms on the edge-stepped defects (**Figure R7**). Meanwhile, the WS₂ superstructure exhibits the edge-rich characteristic, as evidenced by **Figure R8**, which can provide abundant unsaturated sulfur atoms as catalytic active sites for HER.

Finally, the 3D WS₂ superstructure provided an unobstructed mass diffusion pathway to ensure enough reactants reaching the catalytic active sites and in-time release of as-obtained intermediates and productions. Besides, the unique flexible WS₂ superstructure can effectively withstand the impact of high-density gas-liquid exchanges and facilitate mass transfer, endowing exceptional long-term durability under industrial-scale current density.

Through a combination of highly active and abundant catalytic sites as well as efficient mass transfer, the WS₂ superstructure exhibited remarkable and stable HER performance at high current density in the AEM electrolyzer.

Once again, we sincerely express our gratitude to you for all the constructive comments and suggestions, which really help us to make a compelling scientific case that could connect the structural characterizations and electrochemical findings of WS₂ superstructure. The reply for each question/comment is expected to reach the high criteria for you.

Reply to Reviewer 3 and revisions made accordingly:

This manuscript reports the development of flexible WS₂ superstructure catalyst and determined the HER activity. The electrocatalytic performances are really impressive. In addition, the authors have performed an AEM water electrolyzer with impressive cell voltage and stability. However, there are some shortcomings in the work. The work can be considered after the addressing the following queries, which the authors may find is useful.

Response: Great thanks for your supportive feedback on our manuscript. We appreciate your time and effort in reviewing our work and providing professional and invaluable suggestions. As you are concerned, there are some shortcomings that need to be addressed. We have followed your suggestions to further improve the quality of our manuscript. The detailed corrections are listed below.

1. The authors are recommended to add more discussion about the 1T/2H phase of WS₂ and how they are structurally different from the superstructure.

Response: Thanks for your comments and suggestions. Superstructure is rationally designed hierarchical functional structures made of tailored building blocks, which are composed of one or more constituent bulk basic units, leading to effective medium properties beyond those of their ingredients (*Nat. Rev. Chem.* **2022**, *6*, 125–145; *Nat. Rev. Phys.* **2019**, *1*, 198–210; *Science* **2018**, *362*, 808–813; *Chem* **2020**, *6*, 460–471). In our designed WS₂ superstructure (**Figure R18a**), the dangling bond-free nanosheets are staggered with each other in a direction-dependent manner to compose the caterpillar-like three-dimensional (3D) superstructure. This article focuses on the 3D superstructure, as the atomic phase structures of 1T and 2H structures (**Figure R18b**) are a subset of the 3D superstructure.

Figure R18. Conceptual diagram of (a) mesoscopic three-dimensional (3D) superstructure and (b) atomic phase structures (1T and 2H) in our work.

According to your suggestion, we have added more discussion about the 1T/2H phase of WS₂ superstructure based on the XPS and Raman spectra. The tungsten signal is sensitive to its oxidation state and coordination geometry, thus monitoring the position of the binding energy of the W 4f_{7/2} and W 4f_{5/2} core level peaks allows one to unambiguously distinguish the distinct W species and can be used to determine the relative ratio of 1T and 2H phases in the WS₂ superstructure (*Nat. Commun.* **2021**, *12*, 709; *Angew. Chem. Int. Ed.* **2024**, *63*, e202316306). As shown in **Figure R19**, double peaks located at 31.8 eV and 33.8 eV are ascribed to the core levels of W 4f_{7/2} and W 4f_{5/2} of 1T phase WS₂ in the sample, respectively. Two peaks of WS₂ superstructure at 32.7 eV (W 4f_{7/2}) and 34.7 eV (W 4f_{5/2}) are the characteristics of W for 2H phase WS₂ (*Nano Energy*, **2018**, *50*, 176–181; *J. Am. Chem. Soc.* **2022**, *144*, 4863–4873). Phase percentages were calculated by peak area ratios of W 4f and S 2p regions using deconvolution method. As shown in **Figure R19a**, the relative ratio of 1T phase and 2H phase occupies 82.1% and 17.9% in WS₂ superstructure sample, respectively. The

similar results were acquired from the research of S 2p core level spectra (**Figure R19b**).

Raman spectroscopy measurements were also performed to further confirm the phase classification. **Figure R20** showed comparison of typical Raman spectra of fresh WS₂ superstructure, 1T-WS₂ nanosheets, and 2H-WS₂ nanosheets samples. Two prominent peaks corresponding to the in-plane E_{2g}^1 and out-of-plane A_{1g} modes of 2H-WS₂ phase were observed in WS₂ superstructure. The WS₂ superstructure sample exhibited small peaks in the lower frequency region that correspond to the 1T-WS₂ Raman active modes which were not allowed in the 2H-WS₂. These peaks should be attributed to the J_1 , J_2 , and J_3 vibration modes of S-W-S bonds in 1T-WS₂, suggesting WS₂ exists in hybrid 2H and 1T structures in the as-prepared superstructure (*Adv. Mater.* **2015**, *27*, 4837–4844; *Nat. Mater.* **2018**, *17*, 1108–1114).

Figure R19. High-resolution XPS spectra of W 4f (a) and S 2p (b) core level peak regions of the WS₂ superstructure. The fitting red and blue curves represent the contributions of 1T and 2H phases, respectively.

Figure R20. Raman spectra of as-obtained WS₂ superstructure, 1T-WS₂ and 2H-WS₂ nanosheets samples, respectively.

Some necessary discussion has been inserted into our revised manuscript, see page 12 lines 218–228. For your convenience, the related corrections are as follows:

“As shown in Supplementary Fig. 6, double peaks located at 31.8 eV and 33.8 eV are ascribed to the core levels of W 4f_{7/2} and W 4f_{5/2} of 1T phase WS₂ in the sample, respectively. Two peaks of WS₂ superstructure at 32.7 eV (W 4f_{7/2}) and 34.7 eV (W 4f_{5/2}) are the characteristics of W for 2H phase WS₂^{22,23}. Phase percentages were calculated by peak area ratios of W 4f and S 2p regions using the deconvolution method²³. The relative ratio of 1T phase and 2H phase occupies 82.1% and 17.9% in the WS₂ superstructure sample, respectively. Similar results were acquired from the research of S 2p core level spectra. These results revealed that WS₂ existed in hybrid 2H and 1T structures in the as-prepared superstructure. Identical conclusion was obtained in the Raman spectroscopy analysis (Supplementary Fig. 7).”

[22] Voiry, D. et al. Enhanced catalytic activity in strained chemically exfoliated WS₂ nanosheets for hydrogen evolution. *Nat. Mater.* **12**, 850-855 (2013).

[23] Xie, L., Wang, L., Zhao, W., Liu, S., Huang, W., Zhao, Q. WS₂ moiré superlattices derived from mechanical flexibility for hydrogen evolution reaction. *Nat.*

Commun. **12**, 5070 (2021).

2. What is the role of octylamine in the synthesis of WS₂ superstructure and what is the reason to specifically choose octylamine instead of simple amines? Do these amines play any roles in the morphology of the material.

Response: Thanks for your valuable comments. We appreciate the valuable comments and are pleased to clarify these issues. In our reaction, octylamine plays a pivotal role in achieving WS₂ superstructure. It serves not only as a solvent and a reducing agent, but also as an efficient capping agent to the WS₂ superstructure. During the solvothermal synthesis, the octylamine molecules capped on the surfaces of the WS₂ nanocrystallites can effectively suppress the orientation growth of the primary WS₂ nanocrystallites, thus resulting in the formation of disordered quasi-periodic structures and defects on the basal planes (*Nano Energy* **2016**, *22*, 27–37; *Adv. Mater.* **2015**, *27*, 3687–3695). On the other hand, the capped octylamine molecules and some ammonium ions released from the octylamine during the solvothermal process work as the intercalation species, leading to wide interlayer expansion of two adjacent WS₂ monolayers ($\Delta d = 0.34\text{--}0.40$ nm). Consequently, the intercalation-induced interlayer expansion introduces additional lattice defects via lateral disorder, providing more abundant stepped edge surface-terminated structure.

In this article, we designed our synthesis approach around the reaction paths offered by sulfur-amine chemistry. We use a solution of elemental sulfur in amines as reaction precursors because it is less harmful and less sensitive to ambient conditions compared to alternatives such as dialkyldithiocarbamates, carbon disulfide (CS₂), and thioacetamide. (*J. Am. Chem. Soc.* **2003**, *125*, 11100–11105; *J. Mater. Chem.* **2004**, *14*, 2790–2794; *Sci. Rep.* **2018**, *8*, 6591; *Thin Solid Films* **1998**, *315*, 57–61; *Mater. Res. Bull.* **1998**, *33*, 1083–1086; *J. Solid State Chem.* **2002**, *168*, 259–262). Moreover, it is advantageous to utilize cheap, abundant, and relatively stable materials as precursors to lower the cost and the environmental risks related to the manufacture and disposal of industrially electrolyzer-relevant electrocatalysts (*Science* **2023**, *380*, 609–616).

In the current synthesis method, amines are used both as the reaction medium and as the ligands during the formation of the nanocrystals. In the selection of amine species, it has been reported that the solutions of sulfur in tertiary amines were lightly colored or colorless and had a significantly lower conductivity than their primary and secondary counterparts (*J. Am. Chem. Soc.* **1962**, *84*, 2085–2090). An observation of free radicals in colored solutions of sulfur in various amines by electron spin resonance has been reported (*J. Am. Chem. Soc.* **1963**, *85*, 543–546). Primary amines yielded the highest concentration of unpaired electrons, $\geq 10^{-4}$ (1 per 10^4 sulfur atoms), the secondary amines produced less ($\sim 10^{-5}$), and tertiary and some aromatic amines had undetectable amounts ($< 10^{-6}$) of unpaired electrons in solution. It has been reported that the primary and secondary amines open S₈ sulfur rings yielding colored and reactive forms of sulfur, while very pure tertiary amines do not act that way. In addition, the evolution of hydrogen sulfide upon dissolution of sulfur was reported to occur in a variety of short-chain amines (up to n-butylamine) which had a methylene group adjacent to the nitrogen atom (further referred to as α -CH₂).

Bearing these points in mind, we could choose the n-alkyl amines with medium/long chain lengths as ligands, starting from 4-carbon n-butylamine up to 8-carbon octylamine. We performed the synthesis of the WS₂ superstructure using amines with different chain lengths (N-butylamine, N-hexylamine, and N-octylamine) to probe the effect of the type of amine. For all amines utilized in this study, the caterpillar-like WS₂ superstructure could be obtained (**Figure R21**). Scanning electron microscopy (SEM) images indicated that the lateral size of the superstructure can be reduced by employing longer chain amines, thanks to the stronger steric hindrance effect of the longer carbon backbone (*J. Colloid Interface Sci.* **2007**, *310*, 163–166; *Nano Energy* **2016**, *22*, 27–37; *Nat. Commun.* **2012**, *3*, 1177).

Although it is possible to perform synthesis with all the ligands listed above, we found that n-octylamine was the best among the amines utilized in this study for the synthesis of WS₂ superstructure. This choice was made thanks to the combined advantages of commercial availability, low cost, low evaporation rate, and suitable

viscosity of n-octylamine at room temperature, which makes this ligand less costly and easier to work with (Table R8). For this reason, we specifically chose octylamine instead of other simple amines to synthesize the superstructure in our work.

Figure R21. SEM images showing the shapes of superstructure synthesized using different amines. **a**, n-octylamine. **b**, n-hexylamine. **c**, n-butylamine.

Table R8. Boiling point, vapor pressure and price of n-butylamine, n-hexylamine, and n-octylamine.

Ligand	Boiling Point (°C)	Vapor Pressure (mmHg)	Cost (US\$ mL ⁻¹)
N-butylamine (99 %)	78	68 @ 20°C	0.016
N-hexylamine (99 %)	131	7.95 @ 20°C	0.268
N-octylamine (99 %)	175	1 @ 20°C	0.074

3. The PXRD of the as-synthesized material is bit noisy, it is recommended to perform again. In addition, the PXRD patterns of individual 1T and 2H phases also need to be compared with the superstructure.

Response: Thanks for your kind suggestion. Following this suggestion, we have performed the new XRD characterization of the as-synthesized material. In addition, the PXRD patterns of individual 1T and 2H phases have been added to the XRD

analysis for comparison with the superstructure (**Figure R22**).

Figure R22. XRD patterns of as-obtained WS₂ superstructure, 1T-WS₂, and 2H-WS₂ samples. PDF#08-0237 was used as a reference.

We have added the **Figure R22** as a new **Figure 2d** in the revised manuscript. The relevant texts have been added into the revised manuscript (page 12, line 209).

4. What is the reason behind the shifting of 002 planes at lower angles? Is there intercalation of any foreign moiety? Authors may try FTIR study to confirm that.

Response: Thanks for your professional opinions. We appreciate the valuable comments and are pleased to clarify this issue. As shown in **Figure R23**, the (002) diffraction peak of the WS₂ superstructure shifted to 8.9° compared with the 1T-WS₂ and 2H-WS₂ samples, corresponding to an expanded layer spacing of 0.98 nm. The interlayer expansion is due to the intercalation of octylamine molecules during the solvothermal synthesis.

Following your suggestion, we have performed the Fourier transform infrared (FTIR) study to confirm this proposition. FTIR spectra of the as-prepared sample and the pure octylamine were characterized and presented in **Figure R24**. The strong peaks at 2920, 2850, 1430, and 1371 cm⁻¹ are attributed to the vibration of CH₂ and CH₃ groups, and confirms the existence of octylamine molecules in the as-prepared WS₂ superstructure.

The strong peaks from -NH_2 group at 1610 cm^{-1} in the as-prepared WS_2 superstructure are also observed, which may further suggest the intercalation of octylamine molecules between WS_2 layers (*Nano Energy* **2016**, *22*, 27–37; *Nat. Commun.* **2012**, *3*, 1177).

Figure R23. XRD patterns of as-obtained WS_2 superstructure, 1T- WS_2 , and 2H- WS_2 samples. PDF#08-0237 was used as a reference.

Figure R24. FTIR spectra of the as-prepared WS_2 superstructure and the pure octylamine.

We have added the **Figure R24** as **Supplementary Figure 5** in the revised Supplementary Information. The relevant texts have been also added into the revised manuscript (page 12, line 211–214) and revised Supplementary Information (page 6).

5. What is the reason for caterpillar-like morphology displaying better “mechanical flexibility” compared to other available morphologies.

Response: Thanks very much for the valuable comment. As indicated in the previous studies, WS₂ nanomaterials show diverse morphologies mainly including nanosheets, nanotubes, nanospheres, nanowires, nanorods and nanoflowers (*Nat. Mater.* **2013**, *12*, 850–855; *J. Am. Chem. Soc.* **2014**, *136*, 14121–14127; *Adv. Mater.* **2015**, *27*, 4837–4844; *Small* **2017**, *13*, 1603706; *Adv. Mater.* **2018**, *30*, 1705509; *Tungsten* **2020**, *2*, 109–133; *Angew. Chem. Int. Ed.* **2021**, *60*, 21550; *Nat. Commun.* **2021**, *12*, 709; *Sci. Adv.* **2023**, *9*, eadd6167). These WS₂ morphologies generally exhibit in-plane covalent bonding and interlayer van der Waals (VDW) interactions. Although the latter is much weaker than the former, the accumulated forces over the large contact area are still too strong for free interlayer sliding motion, leading to a global brittleness in layered solids.

Figure R25. (a) Schematic diagram of interlayer expansion of 2D layered material. (b) Schematic diagram of WS₂ superstructure based on low Young's modulus nanosheets. (c) Dynamic deformation of flexible WS₂ superstructure in response to shear forces.

In our designed caterpillar-like WS₂ superstructure, through the expansion of the layer spacing of 2D sheets by the intercalation of octylamine molecules, the interlayer

VDW interactions experience a rapid decay, consequently further enabling the activation of relative motion between the sheets (**Figure R25a**). More importantly, the dangling bond-free nanosheets are staggered butted up against each other to establish broad-area plane-to-plane VDW contacts, opening sliding and rotation degrees of freedom between neighboring nanosheets to endow unusual mechanical flexibility (**Figure R25b, c**).

6. It is highly recommended to mention and include more discussion in the main text about the developed phases (1T/2H) of WS₂ superstructure from the XPS spectrum. Further, please mention how superstructure is different from existing 1T/2H phase.

Response: We appreciate you for the helpful suggestions. According to your suggestion, we have added more discussion about the developed phases (1T/2H) of WS₂ superstructure to the main text based on the XPS spectra (**Figure R26**). For your convenience, the added discussions in the main text (page 12, line 218–228) are as follows:

“As shown in Supplementary Fig. 6, double peaks located at 31.8 eV and 33.8 eV are ascribed to the core levels of W 4f_{7/2} and W 4f_{5/2} of 1T phase WS₂ in the sample, respectively. Two peaks of WS₂ superstructure at 32.7 eV (W 4f_{7/2}) and 34.7 eV (W 4f_{5/2}) are the characteristics of W for 2H phase WS₂^{22,23}. Phase percentages were calculated by peak area ratios of W 4f and S 2p regions using the deconvolution method²³. The relative ratio of 1T phase and 2H phase occupies 82.1% and 17.9% in the WS₂ superstructure sample, respectively. Similar results were acquired from the research of S 2p core level spectra. These results revealed that WS₂ existed in hybrid 2H and 1T structures in the as-prepared superstructure. Identical conclusion was obtained in the Raman spectroscopy analysis (Supplementary Fig. 7).”

[22] Voiry, D. et al. Enhanced catalytic activity in strained chemically exfoliated WS₂ nanosheets for hydrogen evolution. *Nat. Mater.* **12**, 850-855 (2013).

[23] Xie, L., Wang, L., Zhao, W., Liu, S., Huang, W., Zhao, Q. WS₂ moiré superlattices

derived from mechanical flexibility for hydrogen evolution reaction. *Nat. Commun.* **12**, 5070 (2021).

Figure R26. High-resolution XPS spectra of W 4f (a) and S 2p (b) core level peak regions of the WS₂ superstructure. The fitting red and blue curves represent the contributions of 1T and 2H phases, respectively.

Superstructures are rationally designed hierarchical functional structures made of tailored building blocks, which are composed of one or more constituent bulk basic units, leading to effective medium properties beyond those of their ingredients (*Nat. Rev. Chem.* **2022**, *6*, 125–145; *Nat. Rev. Phys.* **2019**, *1*, 198–210; *Science* **2018**, *362*, 808–813; *Chem* **2020**, *6*, 460–471). In our designed WS₂ superstructure (**Figure R27a**), the dangling bond-free nanosheets are staggered with each other in a direction-dependent manner to compose the caterpillar-like 3D superstructure. This article focuses on the 3D superstructure, as the atomic phase structures of 1T and 2H structures (**Figure R27b**) are a subset of the 3D superstructure.

Figure R27. Conceptual diagram of (a) mesoscopic three-dimensional (3D) superstructure and (b) atomic phase structures (1T and 2H) in our work.

7. The author should add more discussion on how the defect structure of WS₂ improves the overall activity of the WS₂ superstructure.

Response: We sincerely thank you for the helpful suggestion. Both theoretical (*J. Am. Chem. Soc.* **2005**, *127*, 5308–5309) and experimental (*Science* **2007**, *317*, 100–102; *Nat. Chem.* **2018**, *10*, 1246–1251) studies concluded that the HER activity arises from the sites located along the edges of the two-dimensional (2D) transition metal dichalcogenides (TMDs) layers, while the basal surfaces are catalytically inert. That is, the unsaturated sulfur atoms on the edges play a crucial role in HER catalysis. In our work, the introduction of edge-stepped defects in WS₂ superstructure could expose a large number of W-S dangling bonds and unsaturated sulfur atoms as the active sites for HER.

Capturing direct spectroscopic evidence of intermediates produced during catalytic processes is key to unraveling the origin of HER activity enhancement. (*Nat. Energy* **2019**, *4*, 60–67; *Nat. Commun.* **2016**, *7*, 12440). Surface-enhanced Raman

spectroscopy (SERS) can provide surface-sensitive as well as chemical bond specific signals at the atomic level, making it a powerful fingerprint spectroscopy which can in-situ identify the active sites as well as the surface reaction intermediates during catalytic processes (*Nature* **2021**, *600*, 81–85; *Nat. Commun.* **2023**, *14*, 5289; *Nat. Mater.* **2019**, *18*, 697–701). To reveal how the defect structure of WS₂ improves the overall activity of the WS₂ superstructure, in-situ SERS was used to monitor the HER process in 1.0 M KOH electrolyte.

Figure R28. In-situ SERS spectra of the WS₂ superstructure catalyst at various potentials (vs. RHE) under HER conditions in 1.0 M KOH.

Specifically relevant to the alkaline solution of the HER, where proton generation through water decomposition is pivotal, the assessment of water adsorption capacity assumes critical significance as an indicator of catalytic activity (*Adv. Mater.* **2022**, *34*, 2110604; *Angew. Chem. Int. Ed.* **2024**, *63*, e202316306). As shown in **Figure R28**, the broad peak spanning from 3000 to 3700 cm⁻¹ corresponds to the adsorbed water peaks in the WS₂ superstructure sample, demonstrating its excellent water adsorption capacity (*Adv. Mater.* **2022**, *34*, 2110604; *J. Energy Chem.* **2024**, *92*, 629–638). When the potential reaches -50 mV, an obvious Raman peak located at ca. 2550 cm⁻¹ appears and its intensity increases as the potential further decreases to -200 mV. The band detected

at 2550 cm^{-1} is ascribed to the stretching vibration of S-H bond, $\nu(\text{S-H})$ (*ACS Catal.* **2016**, *6*, 7790–7798; *J. Am. Chem. Soc.* **2020**, *142*, 7161–7167), indicating that the H atom is directly bonded to the sulfur atom of WS₂ superstructure during the HER. Importantly, it is worth mentioning that the vibrational signals, $\nu(\text{W-H})$, between 1831 and 1993 cm^{-1} (*J. Chem. Soc., Dalton Trans.* **1972**, 2492–2496; *J. Phys. Chem. A* **2002**, *106*, 6720–6729) have not been observed in the in-situ SERS measurements (**Figure R28**). Based on detailed fingerprint information mentioned above, it is concluded that the sulfur atom of WS₂ is the catalytic active site for the HER. Albeit different from our WS₂ superstructure, the sulfur atoms in amorphous MoS_x (*ACS Catal.* **2016**, *6*, 7790–7798) and MoS₂ nanosheets (*J. Am. Chem. Soc.* **2020**, *142*, 7161–7167) were also confirmed to be the active sites for HER.

Simultaneously, besides of the HRTEM images (**Figure 2f** and **Supplementary Figure 9**), the irreversible electrochemical oxidation of metal sulfides was also investigated as a measure of their edge sites. Irreversible electrochemical oxidations were performed on both samples (WS₂ superstructure and conventional WS₂ nanosheets, as shown in **Figure R29a**) using 0.5 M H₂SO₄ as the electrolyte at a scan rate of 60 mV s^{-1} . **Figure R29b** shows the cyclic voltammetry (CV) curves of the WS₂ superstructure and conventional WS₂ nanosheets. A peak centered at $\sim 1.05\text{ V}$ (vs. RHE) in the CV curve of conventional WS₂ nanosheets can be ascribed to the oxidation of basal planes of WS₂ (*Faraday Discuss.* **2008**, *140*, 219–231). In contrast, one apparent oxidation peak centered at $\sim 0.67\text{ V}$ vs. RHE was observed in the CV of the WS₂ superstructure, possibly due to its edge-rich (including both sheet edges and stepped defect edges) features, as edges of WS₂ nanostructures are expected to be more readily oxidized than does the basal plane (*Sci. Adv.* **2015**, *1*, e1500259; *ACS Catal.* **2016**, *6*, 861–867). Therefore, the edge-rich characteristic of the WS₂ superstructure is confirmed again.

Figure R29. (a) TEM images of WS₂ superstructure (left) and conventional WS₂ nanosheets (right). The conventional WS₂ nanosheets were synthesized according to a previous report for comparison (*Nat. Commun.* **2021**, *12*, 5070; *J. Am. Chem. Soc.* **2014**, *136*, 14121–14127; *Nano Energy* **2018**, *50*, 176–181). (b) Irreversible electrochemical oxidation cyclic voltammety curves. Irreversible electrochemical oxidation of WS₂ superstructure and conventional WS₂ nanosheets in 0.5 M H₂SO₄ at a scan rate of 60 mV s⁻¹.

To further verify the significance of S active sites for HER, zinc nitrate (Zn(NO₃)₂) was employed in the poisoning experiment to block the S sites, as depicted in **Figure R30**, according to the previous report (*Chem* **2017**, *3*, 122–133; *Angew. Chem. Int. Ed.* **2024**, e202401453; *Nat. Commun.* **2019**, *10*, 5231). As shown in **Figure R31a**, the HER activity of the WS₂ superstructure decreased significantly after being soaked in 1.0 mM

Zn(NO₃)₂ aqueous solution for 30 s. The poisoning of active S sites resulted in an increase of overpotential from 52 mV to 201 mV at 10 mA cm⁻² and 89 to 386 mV at 50 mA cm⁻² (Figure R31b). We can conclude from the poison experiments that edge-stepped defects with abundant unsaturated sulfur atoms play an important role in determining the HER property of the WS₂ superstructure electrocatalyst. During the HER process, these stepped-like structures could provide abundant unsaturated sulfur atoms along with their curved edge profiles to serve as active centers to promote the evolution of H species.

Figure R30. Schematic diagram of the active site blocking mechanism.

Figure R31. HER polarization curves (a) and the required overpotential at 10 and 50 mA cm⁻² (b) of WS₂ superstructure before and after soaking in 1.0 mM Zn(NO₃)₂ solution for 30 s.

We have added the **Figure R29–34** as **Supplementary Figure 22–24** in the revised Supplementary Information. According to your suggestion, we have added more discussion on how the defect structure of WS₂ improves the overall activity of the WS₂ superstructure in the revised manuscript (page 22, lines 387–406). For your convenience, the added discussion are as follows:

“To understand the underlying factors contributing to the excellent activity of the WS₂ superstructure, in-situ surface-enhanced Raman spectroscopy (SERS) was performed to analyze surface reaction intermediates during the HER process⁶⁸. As shown in Supplementary Fig. 22, the broad peak spanning from 3000 to 3700 cm⁻¹ corresponds to the adsorbed water peaks in the WS₂ superstructure sample, demonstrating its excellent water adsorption capacity⁶⁹. This result is in good consistent with the DFT calculations (Fig. 2j). The successful observation of S-H bonds (~2550 cm⁻¹) via in-situ SERS directly confirmed that sulfur atoms serve as the catalytic active sites in WS₂ superstructure for HER^{70–72}. In addition, the irreversible electrochemical oxidation study demonstrated the edge-rich characteristic of the WS₂ superstructure (Supplementary Fig. 23)⁷³. The poison experiments using zinc nitrate further demonstrated that the edge-stepped defects with abundant unsaturated sulfur atoms play an important role in determining the HER property of the WS₂ superstructure electrocatalyst (Supplementary Fig. 24)^{70,74}. We can conclude from the in-situ SERS, irreversible electrochemical oxidation, and poison experiments that edge-stepped defects with abundant unsaturated sulfur atoms play an important role in determining the HER property of the WS₂ superstructure electrocatalyst. During the HER process, these stepped-like structures could provide abundant unsaturated sulfur atoms along with their curved edge profiles to serve as active centers to promote the evolution of H species”

[68] Wang, Y.-H., Zheng, S., Yang, W.-M., Zhou, R.-Y., He, Q.-F., Radjenovic, P., Dong, J.-C., Li, S., Zheng, J., Yang, Z.-L., Attard, G., Pan, F., Tian, Z.-Q., Li, J.-F. In situ Raman spectroscopy reveals the structure and dissociation of interfacial water.

Nature **600**, 81-85 (2021).

- [69] Wang, J., Yang, T., Li, X., Zhang, H., Zhang, Y., He, Y., Xue, H. Hydrogen evolution reaction activity enhancement from active site turnover mechanism. *J. Energy Chem.* **92**, 629-638 (2024).
- [70] Yang, C., Yue, J., Wang, G., Luo, W. Activating and identifying the active site of RuS₂ for alkaline hydrogen oxidation electrocatalysis. *Angew. Chem. Int. Ed.*, e202401453 (2024).
- [71] Chen, J., Liu, G., Zhu, Y.-z., Su, M., Yin, P., Wu, X.-j., Lu, Q., Tan, C., Zhao, M., Liu, Z., Yang, W., Li, H., Nam, G.-H., Zhang, L., Chen, Z., Huang, X., Radjenovic, P.M., Huang, W., Tian, Z.-q., Li, J.-f., Zhang, H. Ag@MoS₂ core-shell heterostructure as SERS platform to reveal the hydrogen evolution active sites of single-layer MoS₂. *J. Am. Chem. Soc.* **142**, 7161-7167 (2020).
- [72] Guo, S., Li, Y., Tang, S., Zhang, Y., Li, X., Sobrido, A.J., Titirici, M.M., Wei, B. Monitoring hydrogen evolution reaction intermediates of transition metal dichalcogenides via operando Raman spectroscopy. *Adv. Funct. Mater.* **30**, 2003035 (2020).
- [73] Miao, J., Xiao, F.-X., Yang, H.B., Khoo, S.Y., Chen, J., Fan, Z., Hsu, Y.-Y., Chen, H.M., Zhang, H., Liu, B. Hierarchical Ni-Mo-S nanosheets on carbon fiber cloth: A flexible electrode for efficient hydrogen generation in neutral electrolyte. *Sci. adv.* **1**, e1500259 (2015).
- [74] Hu, C., Ma, Q., Hung, S.-F., Chen, Z.-N., Ou, D., Ren, B., Chen, H.M., Fu, G., Zheng, N. In situ electrochemical production of ultrathin nickel nanosheets for hydrogen evolution electrocatalysis. *Chem* **3**, 122-133 (2017).

8. Please mention how the Tafel slope is estimated. If it is calculated from dynamic LSV, then the author should consider the backward LSV with 100% iR correction.

Response: We thank you for raising the concern regarding the Tafel slope and for the professional suggestion. All polarization curves were corrected by 100% iR compensation based on measured electrolyte resistance (**Figure R32a**). The

polarization curves were replotted as overpotential (η) versus log current ($\log J$) to get Tafel plots for assessing the HER kinetics of investigated catalysts (**Figure R32b**). By fitting the linear portion of the Tafel plots to the Tafel equation ($\eta = b \log(J) + a$), the Tafel slope (b) can be obtained.

Figure R32. (a) Polarization curves at low current density. (b) Corresponding Tafel slopes derived from (a).

We have added the **Figure R32** as a new **Supplementary Figure 15** in the revised Supplementary Information, and all relevant data have been corrected in the revised manuscript (page 6, line 106–108; page 20, line 347–349).

9. The authors have used Ag/AgCl as a reference electrode, which is highly unstable during long-term stability tests. Therefore, it is recommended to perform a stability study using Hg/HgO as a reference electrode.

Response: We appreciate your professional suggestion. Following your suggestions, we perform a stability study using Hg/HgO as a reference electrode in the conventional three-electrode system. The chronoamperometric measurements were carried out at different high current densities (500, 1000, and 2000 mA cm⁻²) to evaluate the HER stability of the WS₂ superstructure electrode. As displayed in **Figure R33**, the performance degradation of the WS₂ superstructure electrode was negligible for operation at different high current densities using Hg/HgO as reference electrode,

suggesting its potential as a promising electrocatalyst for industrial applications that facilitate lossless transitions across multiple scenarios.

Figure R33. Chronoamperometric (i-t) curves of the WS_2 superstructure at various current densities ($500, 1000, 2000 \text{ mA cm}^{-2}$).

We have added the **Figure R33** as a new **Figure 4e** in the revised manuscript. The relevant texts have been added into the Electrochemical characterization in the revised manuscript (page 34, line 597–598).

10. Please include the fitting circuit and the corresponding fitting parameters of the corresponding EIS plot in Figure 4d. In addition, also mention the potential at which the EIS was performed.

Response: Thanks for the comments and suggestions. We have included the fitting circuit and the corresponding fitting parameters of the corresponding EIS plot in Figure 4d (**Figure R34**). Electrochemical impedance spectroscopy (EIS) test was conducted from 100 kHz to 0.1 Hz at -50 mV vs. RHE , and the alternating current amplitude was 5 mV. The detailed values of all fitting parameters are listed in **Table R9**.

The above-mentioned Figure 4d has been revised as follows:

Figure R34. Nyquist plots for EIS measurements of WS₂ superstructure, 1T-WS₂ nanosheets, and 2H-WS₂ nanosheets, using the frequency in the range from 100 kHz to 0.1 Hz at -50 mV (vs. RHE). The inset is the equivalent circuit model that contains the electrolyte resistance (R_s), constant phase element (CPE) and charge-transfer resistance (R_{ct}). Z' is the real impedance and Z'' is the imaginary impedance.

Table R9. Impedance parameters for the equivalent circuit that was shown in **Figure R34**.

Samples	R_s	CPE-T	CPE-P	R_{ct}
WS ₂ superstructure	0.486	1.08×10^{-3}	0.77	3.87
1T-WS ₂ nanosheets	0.562	4.01×10^{-4}	0.80	8.16
2H-WS ₂ nanosheets	0.697	5.33×10^{-4}	0.76	17.14

We have added the **Figure R34** as a new **Figure 4d** in the revised manuscript and added the **Table R9** as **Supplementary Table 6** in the revised Supplementary Information.

11. The authors mentioned that the superstructure exhibits the “Cutting Edge effect”; could the author clarify this in detail.

Response: Thanks very much for this kind comment. We are pleased to clarify the “Cutting Edge effect” of the superstructure in detail. Gas-involving electrochemical reactions, including gas-evolution reactions (GERs) and gas-consumption reactions (GCRs), are essential components of the energy conversion processes. Generally, GERs include hydrogen evolution reaction (HER), oxygen evolution reaction (OER), hydrazine oxidation reaction (HzOR with N_2 as product), and chlorine evolution reaction (CIER). In water splitting industry, a great number of gas bubbles (H_2/O_2) will generate on the electrode surface as large current densities are usually required. In this case, if the electrode shows strong adhesion to gas bubbles, a large number of gas bubbles will gather around the surface and block the active sites and the diffusion of electrolyte, resulting in huge reaction resistance, as shown in **Figure R35** (*J. Am. Chem. Soc.* **2017**, *139*, 12402–12405; *Accounts Chem. Res.* **2018**, *51*, 1590–1598; *J. Am. Chem. Soc.* **2020**, *142*, 1857–1863; *Adv. Mater.* **2024**, *36*, 2307925; *Nat. Commun.* **2024**, *15*, 2346).

Figure R35. Large number of big H_2 bubbles are generated from the electrodes and gather around surface during HER.

Since the diameters of bubbles released underwater are correlated with the interaction at the three-phase (solid-liquid-gas) contact line (TPCL, as shown in **Figure R36a**), the solid electrode surface intended for promoting the gas evolution in GERs should make

the gas bubbles smaller for faster bubble release (*Adv. Mater.* **2024**, *36*, 2307925; *Nat. Commun.* **2024**, *15*, 2346). The simplified stress analysis on an individual bubble at the electrode surface indicated that two main forces governed the bubble detachment (as shown in **Figure R36b**): buoyant force (F_b) pointing upward and adhesion force (F_a) pointing downward. Thus, the threshold to drive the bubble off the surface can be reached as the buoyant force balances with the adhesion force. The threshold releasing radius is proportional to the root of surface tension ($\sqrt{\gamma}$) and sine function of bubble contact angle ($\sin \alpha$).

$$r \propto \sqrt{\gamma} \sin \alpha$$

The adhesion force of a given course surface (F_{a^*}) is determined by the area fraction of the solid on the surface (f_s).

$$F_{a^*} = f_s F_a$$

where α , r , and γ represent the bubble contact angle, radius of the bubble and the surface tension at the TPCL, respectively. Moreover, F_{a^*} is the adhesion force on rough solid surface.

Figure R36. (a) Continuous three-phase (solid-liquid-gas) contact line (TPCL) (yellow line) at the electrode surface. (b) Stress analysis of one single bubble on the electrode surface.

If the electrode surface is a relatively flat surface, the TPCL should be a continuous circle (**Figure R37a**) and each point of the circle generates the adhesion force. However, the superstructure could break the circle into discontinuous dots where the gaps

between isolated standing units are filled with liquid (**Figure R37b**), named the “cutting edge effect”, resulting in much smaller bubble adhesion force. Therefore, the bubbles on the superstructure exhibit lower adhesion forces and smaller detachment volumes than flat ones; in other words, the superstructure electrodes possess a superior ability to accelerate gas evolution behavior and improve electrocatalytic performance at high reaction rates.

Figure R37. Schematic illustration of how the electrode surface morphology affecting the bubble contacts and release. Continuous and discontinuous TPCL (yellow line) at flat (a) and superstructure electrode surface (b).

To clarify the “Cutting Edge effect” in detail, we have added the **Figure R36** and **37** as new **Supplementary Figure 26** and **27**, respectively, in the revised Supplementary Information. In addition, some necessary explanations have been inserted into our revised Supplementary Information (page 32–34).

12. The author should perform the high-resolution post-XPS study of the WS₂ superstructure to investigate any possible alteration in the chemical state of the W and S. Also determine the relative ratio of 1T and 2H in the XPS spectra after prolonged stability.

Response: We appreciate you for this insightful and constructive recommendation. We have performed the high-resolution post-XPS study of the WS₂ superstructure to investigate any possible alteration in the chemical state of the W and S. As for the W 4f spectra (**Figure R38a**), the signals corresponding to the W⁴⁺ valence states in WS₂ superstructure showed no obvious shifts in binding energies after HER operation in the AEM electrolyzer. This result indicated that the W valence state in WS₂ superstructure was relatively stable after AEM electrolyzer operation. Similarly, the S 2p XPS signals in **Figure R38b** demonstrated that the S chemical state in WS₂ superstructure was relatively stable after AEM electrolyzer operation.

Figure R38. XPS analysis of the WS₂ superstructure anode catalyst before and after AEM electrolyzer operation. (a) W 4f and (b) S 2p XPS signals of the WS₂ superstructure anode catalyst before and after prolonged stability test.

The tungsten signal is sensitive to its oxidation state and coordination geometry, thus monitoring the position of the binding energy of the W 4f_{7/2} and W 4f_{5/2} core level peaks allows one to unambiguously distinguish the distinct W species and can be used to determine the relative ratio of 1T and 2H phases in the WS₂ superstructure (*Nat. Commun.* **2021**, *12*, 709; *Angew. Chem. Int. Ed.* **2024**, *63*, e202316306).

As shown in **Figure R39**, double peaks located at 31.8 eV and 33.8 eV are ascribed to the core levels of W 4f_{7/2} and W 4f_{5/2} of 1T phase WS₂ in the sample, respectively. Two peaks of WS₂ superstructure at 32.7 eV (W 4f_{7/2}) and 34.7 eV (W 4f_{5/2}) are the

characteristics of W for 2H phase WS₂ (*Nano Energy*, **2018**, *50*, 176–181; *J. Am. Chem. Soc.* **2022**, *144*, 4863–4873). Phase percentages were calculated by peak area ratios of W 4f and S 2p regions using deconvolution method. As shown in **Figure R39a**, the relative ratio of 1T phase and 2H phase occupies 82.1% and 17.9% in WS₂ superstructure sample before prolonged stability test, respectively. After the prolonged stability test, the post-XPS revealed that the relative ratio of 1T phase and 2H phase occupies 81.0% and 19.0% in WS₂ superstructure sample (**Figure R39b**). The similar results were acquired from the research of S 2p core level spectra (**Figure R39**). These results indicated that the chemical state of the W and S in WS₂ superstructure were relatively stable after prolonged stability test.

Figure R39. High-resolution XPS spectra of W 4f (left) and S 2p (right) core level peak regions of the WS₂ superstructure anode catalyst before (a) and after (b) prolonged stability test. The fitting red and blue curves represent the contributions of 1T and 2H phases, respectively.

We have added the **Figure R38** and **39** as new **Supplementary Figure 36** and **37**, respectively, in revised Supplementary Information. In addition, some necessary explanations have been inserted into our revised Supplementary Information (page 44–45).

Once again, we would like to express our sincere thanks to you for the insightful comments and kind suggestions regarding the superstructure synthesis, defect and phase structure, and electrochemical testing sections! We hope that the responses to each question/comment meet your high standards.

REVIEWERS' COMMENTS

Reviewer #1 (Remarks to the Author):

The authors have responded to the queries raised by the reviewers satisfactorily. The manuscript can be considered for publication in its current stage.

Reviewer #3 (Remarks to the Author):

Authors have very elaborately revised the manuscript by considered every query of the reviewers. They have address each point very meticulously and the suggested experiments have been performed and the results are included in the manuscript and discussed in length. The revised manuscript is suitable for consideration for publication in Nat Commun.

Point-by-point response to the reviewers' comments

Dear Reviewers,

Thanks for your valuable time to constructive comments on our manuscript titled “Flexible tungsten disulfide superstructure engineering for ultrastable alkaline hydrogen evolution in anion exchange membrane water electrolyzers” for *Nature Communications* (NCOMMS-24-00276A).

Reviewer #1 (Remarks to the Author): The authors have responded to the queries raised by the reviewers satisfactorily. The manuscript can be considered for publication in its current stage.

Response: Thank you for agreeing to publish our manuscript. We also sincerely appreciate your comments and suggestions on our work, which have certainly improved our manuscript.

Reviewer #3 (Remarks to the Author): Authors have very elaborately revised the manuscript by considered every query of the reviewers. They have address each point very meticulously and the suggested experiments have been performed and the results are included in the manuscript and discussed in length. The revised manuscript is suitable for consideration for publication in Nat. Commun.

Response: We thank you for the recommendation for the publication of our manuscript. Thanks for your positive evaluation of this work and valuable suggestion again.